# Coupled Transformer Autoencoder for Disentangling Multi-Region Neural Latent Dynamics

**Ram Dyuthi Sristi**
UC San Diego
rsristi@ucsd.edu

**Sowmya Manojna Narasimha**
UC San Diego
smnarasimha@ucsd.edu

**Jingya Huang**
UC San Diego
jih201.ucsd@gmail.com

**Alice Despatin**
RWTH Aachen University & FZ Jülich
alice.despatin@rwth-aachen.de

**Simon Musall**
RWTH Aachen University & FZ Jülich
s.musall@fz-juelich.de

**Vikash Gilja**
UC San Diego
vgilja@ucsd.edu

**Gal Mishne**
UC San Diego
gmishne@ucsd.edu

## Abstract

Simultaneous recordings from thousands of neurons across multiple brain areas reveal rich mixtures of activity that are shared between regions and dynamics that are unique to each region. Existing alignment or multi-view methods neglect temporal structure, whereas dynamical latent-variable models capture temporal dependencies but are usually restricted to a single area, assume linear read-outs, or conflate shared and private signals. We introduce Coupled Transformer Autoencoder (CTAE)—a sequence model that addresses both (i) non-stationary, non-linear dynamics and (ii) separation of shared versus region-specific structure, in a single framework. CTAE employs Transformer encoders and decoders to capture long-range neural dynamics, and explicitly partitions each region's latent space into orthogonal shared and private subspaces. We demonstrate the effectiveness of CTAE on a controlled synthetic dataset and two high-density electrophysiology datasets of simultaneous recordings from multiple regions, one from motor cortical areas and the other from sensory areas. CTAE extracts meaningful representations that better decode behavior variables compared to existing approaches.

## 1 Introduction

The advent of high-density electrophysiology probes, e.g., Neuropixels, and volumetric calcium imaging has enabled recording large-scale, high-resolution, multi-region neuronal datasets. This shift from single-area recordings to *distributed circuits* reveals both globally coordinated signals and region-specific specialization (Jun et al., 2017; Machado et al., 2022). Individual neurons both receive input from and project to multiple distant areas; behaviorally relevant codes are hypothesized to be broadcast across widespread circuits rather than confined to a canonical locus (Machado et al., 2022). This perspective challenges conclusions drawn from single-area studies and raises the question of which computations are local, which are distributed, and how global brain states are coordinated. Disentangling the *shared* components that mediate inter-area interactions from *private* signals unique to each region is therefore essential—both for mechanistic insight and for designing causal experiments such as targeted optogenetic inactivation or electrical stimulation of an upstream circuit.

Thus, recent work has focused on analyzing *distributed circuits* to recover latent activity patterns both within and across regions. A successful multi-region latent model faces three simultaneous challenges: (i) latent trajectories must evolve smoothly in time to respect neural autocorrelation; (ii) they must accommodate the non-stationary, nonlinear dynamics that real circuits display; and (iii)

they must separate shared from region-specific structure without a parameter explosion as the number of areas grows.

Even within a single brain region, recordings of neural population activity reveal a mixture of structured responses and substantial trial-to-trial variability, introducing a challenge when interrogating the behavior of neural circuits (Cunningham & Yu, 2014). A widely adopted framework—neural latent dynamics (Vyas et al., 2020; Churchland & Shenoy, 2024) —proposes that these high-dimensional responses reflect the evolution of a consistent, low-dimensional trajectory, supported by stable patterns of co-variability across neurons. Classical PCA or factor analysis (FA) uncover low-dimensional structure(Cunningham & Yu, 2014) but ignore time. Moreover, because observed spikes are subject to high Poisson noise, the resulting latents can exhibit high-frequency dynamics that are difficult to interpret and may not reflect meaningful underlying processes. Consequently, latent variable models that incorporate temporal smoothness or dynamical constraints have been developed, based on either linear Gaussian Process (GP) models (Yu et al., 2009) or nonlinear deep learning models (Pandarinath et al., 2018; Ye & Pandarinath, 2021). Yet, when these single-area tools are applied naïvely to multi-area data, e.g., by concatenating the recordings, they often fail. Inter-area delays warp the latent space, differences in correlation structure cause shared factors to absorb private variance, and the larger or more active region can dominate the mixture weights. To address these challenges, recent studies have investigated both predictive models (Zandvakili & Kohn, 2015; Semedo et al., 2019; Perich et al., 2018) and joint latent variable (Gokcen et al., 2022;?) approaches for inter-regional activity. These efforts reveal that only a selective subset of latent dimensions actively participates in inter-area communication. Such findings motivate extensions to the latent dynamics framework, proposing that communication is mediated through a persistent, low-dimensional subspace—referred to as a communication subspace (Semedo et al., 2019)—that is distinct and orthogonal to private, region-specific dynamics Fig. 1.

More recently, joint latent variable models inspired by canonical correlation analysis (CCA) have been developed to capture correlated latent dynamics across regions while simultaneously learning region-specific latent components. One approach focused on neural data has been *linear* approaches that generalize GP-based models to the multi-region setting (Gokcen et al., 2022; 2023; 2024; Li et al., 2024). Alternatively, nonlinear autoencoder-based disentangling methods (Lee & Pavlovic, 2021; Koukuntla et al., 2024) have been developed to analyze multiview point cloud data and partition latents into shared and private factors. However, these treat time points as i.i.d. samples and therefore discard dynamics.

**This work.** We introduce the *Coupled Transformer Autoencoder* (CTAE), an end-to-end framework for modeling simultaneous recordings from multiple brain areas. Transformer-based (Vaswani et al., 2017) encoders and decoders act as flexible priors capable of capturing non-stationary, long-range neural dynamics; each region's latent space is split into *orthogonal shared and private subspaces*, with reconstruction losses preserving region-specific information and a lightweight alignment loss matching the shared representations. Because the latents are behaviour-agnostic, downstream decoding—whether kinematics, forces, or cognitive variables—can be carried out on the same embedding without retraining. Our contributions are as follows:

- **Non-stationary multi-region modelling.** CTAE extracts shared and private representations that capture long-range, nonlinear dynamics absent.
- **Scalable architecture.** A mixing weight allows the shared subspace to extend trivially beyond two regions without an exponential increase in parameters.
- **Generic downstream utility.** The behaviour-agnostic latent space supports diverse decoding tasks (e.g. position, velocity, cognitive state) using simple linear read-outs.
- **Empirical validation.** On neural recordings from M1–PMd (Utah Arrays) and SC-ALM (Neuropixel Probes), CTAE achieves higher shared-variance capture than existing works, while revealing interactions consistent with anatomy.

## 2 RELATED WORK

**Single-region latent-variable models.** Linear techniques—including PCA (Pearson, 1901) and FA (Harman, 1976) are widely used to analyze large-scale recordings (Cunningham & Yu, 2014). Gaussian-Process Factor Analysis (GPFA) adds a smooth GP prior to FA to enforce temporal

continuity (Yu et al., 2009). GPFA's stationarity and linear read-out assumptions are relaxed by Latent Factor Analysis via Dynamical Systems (LFADS) (Pandarinath et al., 2018) and the Neural Data Transformer (NDT) (Ye & Pandarinath, 2021), which exploit nonlinear recurrent generators or self-attention to capture non-stationary and nonlinear dynamics. Subsequent efforts pursued identifiability with switching LDS (Linderman et al., 2016; Glaser et al., 2020), GP-SLDS (Hu et al., 2024), locally linear manifold models in DFINE (Abbaspourazad et al., 2024), and variational neural state-space models such as VIND (Hernandez et al., 2018) and S4-based sequence layers (Gu et al., 2022).

**Multi-region latent models.** Multi-region methods relying on GP models include Delayed Latents Across Groups (DLAG) (Gokcen et al., 2022), its multi-population generalization mDLAG (Gokcen et al., 2023), multi-view GPFA extensions (Gokcen et al., 2024), and Multi-Region Markovian Gaussian Process (Li et al., 2024). These inherit smooth-GP assumptions and linear read-outs that struggle with non-stationary or long-range dependencies. Classical multiset CCA (Hotelling, 1936) and tensor-decomposition methods (Cichocki et al., 2015) have also been applied to multi-area recordings, however these are linear techniques. Current-Based Decomposition (CURBD) (Perich et al., 2020) infers directed currents between many regions from data-constrained RNNs, yet it does not model explicit shared–private split. Similarly, multi-modal models have been recently developed to jointly analyze neural recordings and behavior Vahidi et al. (2025); Gondur et al. (2024); Yi et al. (2025). These models could potentially be adapted to analyze multi-region recordings, however we expect Multimodal GP-VAE Gondur et al. (2024) and Shared-AE Yi et al. (2025) to scale poorly to more than 2 regions due to exponential growth in experts or pairwise loss terms. Both also operate on i.i.d. samples or short windows rather than full neural time series.

**Multiview autoencoders** Multi-view representation learning has progressed beyond classical CCA by incorporating additional losses and constraints to more effectively identify shared latent representations. However, these methods often require adaptation to account for the specific statistical and temporal properties of neural data. Correlation-based alignments—canonical correlation analysis (CCA) and its deep variant (DCCA) (Andrew et al., 2013)—maximise instantaneous correlation but provide no guarantee of capturing *all* shared variance; Deep CCA Autoencoders (DCCAE) add reconstructions at the risk of mixing private with shared information (Wang et al., 2016). Representative examples include Deep Coupled Auto-encoder Networks (Wang et al., 2014), Coupled Autoencoders for domain adaptation (Wang & Breckon, 2023), Correlated Autoencoders for audio-visual retrieval (Feng et al., 2014), Deep Correlation Autoencoders (DCCAE) (Wang et al., 2016), Split-brain Autoencoders (Zhang et al., 2017), shared-private domain adaptation AEs (Bousmalis et al., 2016) and cycle-consistent multiview AEs (Wu et al., 2018). SPLICE (Koukuntla et al., 2024) extends this line with measurement-network penalties to sharpen disentanglement, however it scales poorly—its auxiliary *measurement networks* grow exponentially with region count—while DMVAE assumes one global shared component across *all* views, precluding subset-specific latents. Multimodal VAE (MVAE) (Wu & Goodman, 2018), Disentangled Multimodal VAE (DMVAE) (Lee & Pavlovic, 2021), Joint Multimodal VAE (Sutter et al., 2021), Mixture-of-Experts MVAE (Shi et al., 2019) learn shared/private factors across images, text and audio but treat each sample independently, ignoring sequential dependencies.

Across these categories, no existing method jointly satisfies *(i)* non-stationary nonlinear dynamics, *(ii)* temporal continuity and *(iii)* scalability to more than two regions without incurring a parameter explosion as the number of areas increases.

## 3 PROBLEM FORMULATION

Let $\boldsymbol{X}^{(1)} \in \mathbb{R}^{N_1 \times T}$, $\boldsymbol{X}^{(2)} \in \mathbb{R}^{N_2 \times T}$ denote simultaneous neural population recordings from two brain regions, each acquired over $T$ time steps and, $N_1$ and $N_2$ channels respectively (generalization to more than 2 regions is in Appendix B). We assume that the observed neural activity at each time step $t$ is a nonlinear transformation of latent variables (Fig.2), which include: (i) shared latent dynamics within a communication subspace (Semedo et al., 2019) spanned by neural trajectories that are correlated across regions, and (ii) private latent dynamics that capture region-specific processes lying in subspaces orthogonal to the shared component. To effectively capture the temporal structure

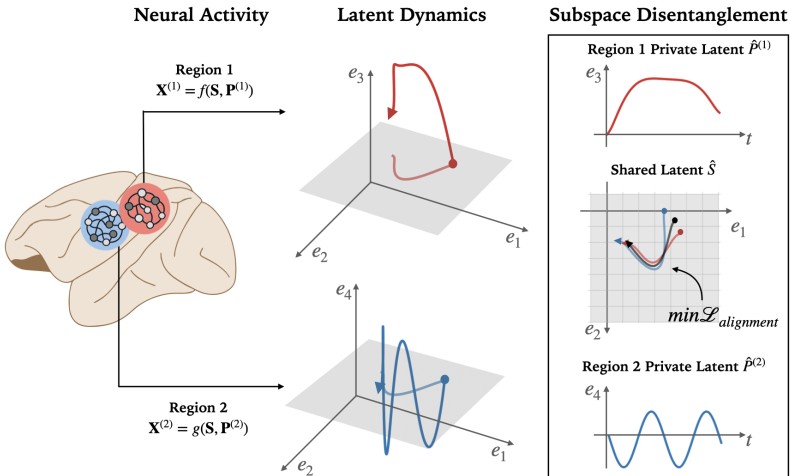

Figure 1: Observed neural activity across time from two brain regions, denoted as $\boldsymbol{X}^{(1)}$ and $\boldsymbol{X}^{(2)}$, is modeled as a nonlinear function of underlying latent dynamics specific to each region. In the illustration, $[e_1, e_2, e_3]$ span the latent subspace for region 1 and $[e_1, e_2, e_4]$ for region 2. Inter-regional communication is mediated by shared latent trajectories $S$ within the common subspace $[e_1, e_2]$, which drive correlated population activity. Following the output-null/potent hypothesis, we assume shared and private dimensions are orthogonal, allowing clear recovery of shared dynamics from region-specific processes.

of neural activity, we model these latent variables as functions of the entire observed neural activity.

$$\mathbf{X}_t^{(1)} = f\big(\boldsymbol{S}_{1:t}, \boldsymbol{P}_{1:t}^{(1)}\big), \qquad \mathbf{X}_t^{(2)} = g\big(\boldsymbol{S}_{1:t}, \boldsymbol{P}_{1:t}^{(2)}\big) \qquad \text{for } t \in \{1, \dots, T\} \tag{1}$$

where $1:t$ denotes the sequence of indices between 1 and $t$, $\boldsymbol{S}_t \in \mathbb{R}^{d_s}$ represents the *shared* dynamics between the two regions at time $t$, and $\boldsymbol{P}_t^{(1)} \in \mathbb{R}^{d_1}$ and $\boldsymbol{P}_t^{(2)} \in \mathbb{R}^{d_2}$ capture dynamics specific to regions 1 and 2, respectively, and $d_s$, $d_1$, and $d_2$ denote the dimensionalities of the shared and private representations.

Our goal is to recover the shared and private latent dynamics $\boldsymbol{S}$, $\boldsymbol{P}^{(1)}$, and $\boldsymbol{P}^{(2)}$ given the observed neural activity $\mathbf{X}^{(1)}$ and $\mathbf{X}^{(2)}$.

**Notations.** Bold upper-case letters denote matrices (e.g., $\mathbf{X}$), bold lower-case denote latent vectors (e.g., $\boldsymbol{w}$), and $\widehat{\cdot}$ indicates reconstructions. Let $\mathbf{1}_n$ denote a vector of all-ones of length $n$ and let $\mathbf{0}_n$ denote a vector of all zeros of length $n$. $\|(\cdot)\|_F$ denotes the Frobenius norm. $\odot$ denotes element-wise multiplication. Expectation is over batches of trials unless stated otherwise.

## 4 Coupled Transformer Autoencoders for Two Regions

Our model is comprised of a coupled autoencoder built on transformer models, that disentangles shared and private representations. We have designed loss functions to recover latents that maximally represent the neural activity, where the shared representations are aligned across all regions, and each disentangled sub-space is orthogonal to the others. For clarity, we describe the two-region case in the main text; the extension to an arbitrary number of regions is provided in Appendix B.

### 4.1 Model Architecture

We design separate causal Transformer-based encoder–decoder pairs, denoted by $(E_\theta^{(1)}, D_\phi^{(1)})$ and $(E_\theta^{(2)}, D_\phi^{(2)})$, for each of the two brain regions. Each encoder is a Transformer stack with *self-attention* layers that capture long-range, nonlinear temporal dependencies within a region. The recorded spike trains are Gaussian-smoothed to obtain continuous firing-rate estimates which are the inputs to the encoder. Each decoder employs standard Transformer *cross-attention* to reconstruct the original

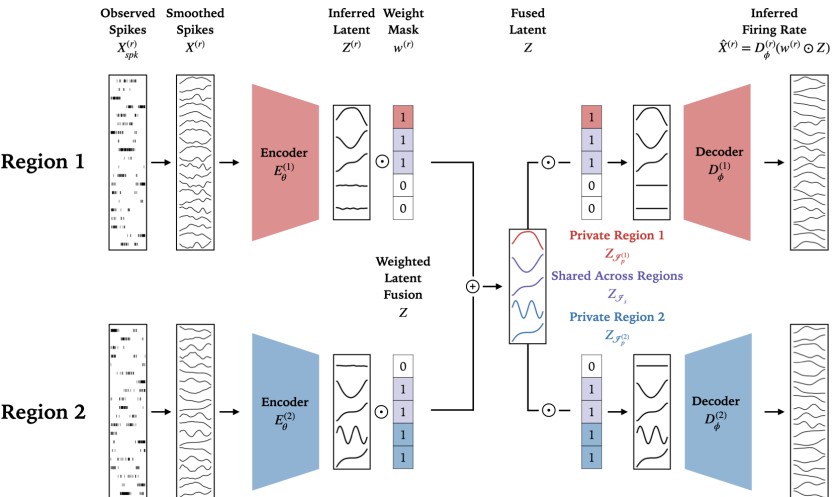

Figure 2: **CTAE architecture.** CTAE is composed of a coupled autoencoder, where the encoders and decoders are causal transformers designed to reconstruct neural activity for each region $r$. The inputs to the network are estimated spike rates from each region. A weight mask per region $w^{(r)}$ is used to disentangle the shared representation (violet) from the region-specific latents (red and blue) in the encoder outputs $Z^{(r)}$. The latents are recovered via end-to-end training.

firing rates from its region's latents. Processing the full multichannel time series from each session, the encoders produce latent representations partitioned into two distinct components: shared latent sequences capturing dynamics common to both regions and private latent sequences specific to each region:

$$\mathbf{Z}^{(1)} = E_\theta^{(1)}(\mathbf{X}^{(1)}), \qquad \mathbf{Z}^{(2)} = E_\theta^{(2)}(\mathbf{X}^{(2)}), \qquad \mathbf{Z}^{(1)}, \mathbf{Z}^{(2)} \in \mathbb{R}^{D \times T} \tag{2}$$

Here $D$ is the total number of latent dimensions. To indicate which of the latent dimensions correspond to shared and private representations, we introduce weight vectors $\mathbf{w}_1$ and $\mathbf{w}_2 \in \{0,1\}^D$.

**Region–specific weight masks.** Let the latent dimension be partitioned as $D = d_s + d_1 + d_2$. Define three contiguous index sets $\mathcal{I}_s = \{1, \ldots, d_s\}$, $\mathcal{I}_1 = \{d_s+1, \ldots, d_s+d_1\}$, and $\mathcal{I}_2 = \{d_s+d_1+1, \ldots, D\}$. We then construct binary weight vectors that indicate which subset of the latent dimensions correspond to the shared or to region-private. Define $\mathbf{w}_1, \mathbf{w}_2 \in \{0,1\}^D$ as

$$\mathbf{w}_1 = \Big[\underbrace{\mathbf{1}_{d_s}}_{\mathcal{I}_s}, \underbrace{\mathbf{1}_{d_1}}_{\mathcal{I}_1}, \underbrace{\mathbf{0}_{d_2}}_{\mathcal{I}_2}\Big]^\top, \qquad \mathbf{w}_2 = \Big[\underbrace{\mathbf{1}_{d_s}}_{\mathcal{I}_s}, \underbrace{\mathbf{0}_{d_1}}_{\mathcal{I}_1}, \underbrace{\mathbf{1}_{d_2}}_{\mathcal{I}_2}\Big]^\top. \tag{3}$$

so that $\mathbf{w}_1$ activates the shared and region-private of region 1 dimensions, whereas $\mathbf{w}_2$ activates the shared and region-private dimensions of region 2. Throughout the paper we treat $\mathbf{w}_r$ as fixed; the latent space dimensions $(d_s, d_1, d_2)$ are treated as hyper-parameters and tuned on a validation set. Importantly, these masks do not hard-code the actual interaction structure: they only specify an *upper bound* on how many shared or private latents the model may allocate. During training, dimensions unsupported by the data naturally collapse to negligible variance. While fixed masks provide clarity and control in the two-region setting, they could also be initialized from anatomical priors or made learnable so that the model itself infers which latents are shared versus private. We leave these scalable extensions to future work.

**Weighted latent fusion.** Broadcasting the masks across time, we aggregate the two latent trajectories dimension-wise via a masked average for each dimension $d$ in the latent:

$$\mathbf{Z}_t[d] = \frac{\mathbf{w}_1[d]\mathbf{Z}_t^{(1)}[d] + \mathbf{w}_2[d]\mathbf{Z}_t^{(2)}[d]}{\mathbf{w}_1[d] + \mathbf{w}_2[d]}. \tag{4}$$

Equation equation 4 leaves the private blocks unchanged (they receive weight 1 from only their own region)

$$\widehat{\boldsymbol{P}}^{(1)} = \mathbf{Z}_{\mathcal{I}_1} = \mathbf{Z}_{\mathcal{I}_1}^{(1)}, \quad \widehat{\boldsymbol{P}}^{(2)} = \mathbf{Z}_{\mathcal{I}_2} = \mathbf{Z}_{\mathcal{I}_2}^{(2)}. \tag{5}$$

while averaging the shared block across regions:

$$\widehat{\boldsymbol{S}} \;=\; \mathbf{Z}_{\mathcal{I}_s} \;=\; \tfrac{1}{2}\big(\mathbf{Z}_{\mathcal{I}_s}^{(1)} + \mathbf{Z}_{\mathcal{I}_s}^{(2)}\big). \tag{6}$$

This design will implicitly force alignment of the shared latents of both regions, via minimizing the reconstruction losses we define in the next section.

**Region-specific decoding.** Each decoder receives only the latent dimensions relevant to its own region:

$$\widehat{\mathbf{X}}^{(r)} \;=\; D_\phi^{(r)}\big((\mathbf{w}_r\mathbf{1}_T^\top)\odot\mathbf{Z}\big), \tag{7}$$

where $(\mathbf{w}_r\mathbf{1}_T^\top) \in \mathbb{R}^{D\times T}$ is an outer product such that each weight is duplicated for each dimension along all time points. Thus, the element-wise product zeroes out unrelated latents, forcing decoder $r$ to rely exclusively on the subset of dynamics meaningful for its region.

This architecture (i) aligns both regions in a common latent space of dimension $D$, (ii) preserves region-specific structure via fixed relevance masks, and (iii) enforces cross-region consistency through the fusion rule in Eq. equation 4. The loss function balancing reconstruction accuracy and alignment objectives is detailed in Section 4.2. We illustrate our architecture in Fig. 2.

## 4.2 TRAINING OBJECTIVE

We use four loss functions to recover the shared and private latent representations with CTAE:

**Reconstruction loss.** The first loss is a reconstruction loss, so that each transformer autoencoder faithfully reproduces its own region's activity:

$$\mathcal{L}_{\text{rec}} \;=\; \sum_{r=1}^{2}\big\|\widehat{\mathbf{X}}^{(r)} - \mathbf{X}^{(r)}\big\|_F^2. \tag{8}$$

**Shared-only reconstruction.** To ensure that *all* structure common to both regions is routed into the shared block, we define a loss such that each decoder reconstructs the neural activity from its region using *only* the shared representations. Let $\mathbf{w}^{(s)} = \mathbf{w}_1\odot\mathbf{w}_2 = [\,\mathbf{1}_{d_s}, \mathbf{0}_{d_1}, \mathbf{0}_{d_2}]^T$ be the intersection mask that selects only the shared dimensions. The loss is

$$\mathcal{L}_{\text{shared}} \;=\; \sum_{r=1}^{2}\big\|D_\phi^{(r)}\big((\mathbf{w}^{(s)}\mathbf{1}_T^\top)\odot\mathbf{Z}\big) - \mathbf{X}^{(r)}\big\|_F^2, \tag{9}$$

where $\mathbf{w}^{(s)}$ zeroes out all private coordinates from the inputs to the decoder, thus forcing the model to encode every cross-region regularity inside the shared subspace. Without this constraint, shared information can shift into private subspaces, leading to inaccurate representation.

**Alignment loss.** The shared latents are meant to capture only dynamics common to both regions. To enforce this, we align each encoder's shared output to their average, ensuring consistency across regions and preventing region-specific variance from leaking into the shared space:

$$\mathcal{L}_{\text{align}} = \sum_{r=1}^{2}\big\|(\mathbf{w}_r\mathbf{1}_T^\top)\odot\mathbf{Z} - (\mathbf{w}_r\mathbf{1}_T^\top)\odot\mathbf{Z}^{(r)}\big\|_F^2. \tag{10}$$

**Orthogonality loss.** To encourage each latent coordinate to capture distinct, non-redundant structure, we penalize correlations between *all* rows of the fused latent matrix $\mathbf{Z} \in \mathbb{R}^{D\times T}$. Let $\mathbf{G} = \frac{1}{T}\mathbf{Z}\mathbf{Z}^\top$ denote the empirical Gram matrix of latent trajectories, whose off-diagonal entries encode row-wise correlations. We drive only the *off-diagonal* entries of $\mathbf{G}$ towards zero,

$$\mathcal{L}_{\text{orth}} \;=\; \big\|\mathbf{G} - \text{diag}(\mathbf{G})\big\|_F^2, \tag{11}$$

so that every pair of latent dimensions—shared or private—becomes approximately orthogonal, promoting global disentanglement.

Finally, the complete objective minimizes the weighted sum of the four losses:

$$\mathcal{L} \;=\; \mathcal{L}_{\text{rec}} + \lambda_{\text{align}}\,\mathcal{L}_{\text{align}} + \lambda_{\text{shared}}\,\mathcal{L}_{\text{shared}} + \lambda_{\text{orth}}\,\mathcal{L}_{\text{orth}}. \tag{12}$$

The weights $\lambda_{\text{shared}}, \lambda_{\text{align}}, \lambda_{\text{orth}}$ are selected on a held-out validation set. Methodology for training the architecture is summarized in Algorithm 1.

# 5 EXPERIMENTS[1]

We analyze two real-world datasets of simultaneous neural recordings from a motor circuit and from a multisensory circuit. In addition, we include a controlled synthetic dataset in Appendix C. We demonstrate the efficacy of CTAE in extracting shared and private dynamics in comparison to DLAG (Gokcen et al., 2022). We also provide a comparison to Deep CCA (Andrew et al., 2013) in Appendix H. Implementation details for each experiment are in the appendix.

## 5.1 MOTOR CIRCUIT: M1-PMD

We applied CTAE to simultaneous neural recordings from dorsal premotor cortex (PMd) and primary motor cortex (M1) in macaque monkeys performing a standard delayed center-out reaching task with eight outward targets (Perich et al., 2018), additional details in Appendix E.1. The dataset consisted of 208 total trials across 8 reach conditions, with spike-sorted data from 66 and 52 putative neurons in PMd and M1, respectively, recorded via 64-channel Utah arrays. Each trial spanned 3s, with spikes and behavioral variables (e.g., position) binned in 100 ms bins, resulting in 30 time points per trial.

Historically, PMd and M1 have been jointly analyzed due to their complementary roles in movement preparation and execution. During the instructed delay period between stimulus onset and the go cue, both regions exhibit preparatory activity without inducing muscle output (Kaufman et al., 2014; Churchland & Shenoy, 2024), potentially representing task goals—i.e., target identity (Byron et al., 2010; Lara et al., 2018). Following the 'go' cue, both PMd and M1 generate execution-related activity that descends to motor neurons and enables decoding of continuous hand kinematics, including position and velocity (Churchland & Shenoy, 2024; Elsayed et al., 2016; Gilja et al., 2012; Byron et al., 2010).

We applied CTAE to the dataset for unified modeling of joint latent subspaces and identified:

- A shared subspace of dimension $d_s$, capturing correlated dynamics across PMd and M1.
- Private subspaces of dimensions $d_{\text{PMd}}$ and $d_{\text{M1}}$, capturing region-specific activity.

Fig. 3 visualizes a subset of the CTAE inferred neural latent dynamics (the set of all latents is in Fig. 14). Each panel shows a distinct latent dimension evolving over time. The shared latent subspace captures both condition-invariant and condition-dependent structure, encoding aspects such as reach direction and temporal progression. Region-specific latents display more complex dynamics, likely reflecting local circuit processes.

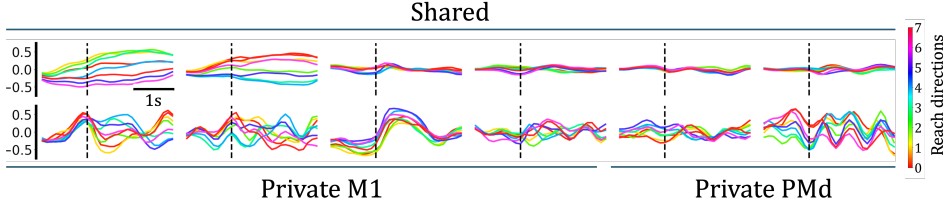

Figure 3: M1-PMd dataset. CTAE shared (top) and region-private (bottom) latents. Dotted vertical line indicates the "go" cue in each trial.

To interpret the latent dynamics—shared $Z^s$, and private $Z^{(\text{PMd})}$, $Z^{(\text{M1})}$, we evaluated the inferred latents using two decoding tasks with simple linear decoders:

**Continuous decoding hand position**: Predict hand position at each time point after go-cue (last 2 seconds for each trial) using linear regression: $v_t^{\text{hand}} = W \mathbf{z}_t$. We plot in Figure 4A the predictions from neural activity, from CTAE latents and from DLAG latents.

**Discrete decoding target condition**: Classify the reach condition (one of 8 targets) using multi-class logistic regression: $\hat{y}_i = \frac{\exp(\mathbf{w}_i^\top \mathbf{z} + b_i)}{\sum_{j=1}^{C} \exp(\mathbf{w}_j^\top \mathbf{z} + b_j)}$ for $i = 1, \ldots, C$. We present in Figure 4B the confusion matrices across conditions for neural activity, CTAE and DLAG latents.

---

[1]Code for CTAE is available in `https://github.com/Mishne-Lab/ctae-multiregion`

Figure 4: M1-PMd dataset. a) Ground truth hand position in top-left corner. Hand position decoding from neural activity in M1 and PMd (top), from CTAE latents (middle) and from DLAG (bottom). b) Confusion matrices for reach direction classification (order of plots is same as in A). Classification accuracy in parentheses.

We hypothesized that the shared latent space would encode the majority of behaviorally relevant information—particularly target identity—while PMd might contribute higher-order planning signals, and M1 reflect finer temporal structure related to movement execution.

Our results confirmed that CTAE's shared latent factors accounted for the majority of variance associated with reaching behavior (Fig. 4 a). In other words, the shared PMd–M1 subspace captured the dominant task-relevant signals. In contrast, DLAG tends to distribute behaviorally relevant variance across private PMd and private M1 latents in a directionally anisotropic manner. This is reflected in both private latents predicting certain movement directions but not others, and also discrete target classification showing high accuracy for only a subset of reach directions. Such direction-specific "fragmentation" of kinematic information across private latents is inconsistent with established findings that both PMd and M1 encode reach kinematics through a coherent, low-dimensional manifold. This pattern suggests that DLAG may inadvertently leak shared information into private subspaces, whereas CTAE's mask structure and regularization prevent such leakage. We also compare to DeepCCA in Appendix H, and to analyzing the concatenation of data from both regions with a single Transformer AE in Appendix I, and demonstrate that both baselines have significantly lower continuous and discrete prediction accuracy than both CTAE and DLAG shared latents. These findings demonstrate that CTAE can simultaneously decode continuous behavioral trajectories Fig. 4 a. and classify discrete behavioral states Fig. 4 b. more effectively than prior approaches.

We present time-wise prediction of the reach in Fig. 19. Importantly, our results support the view that motor planning and execution signals are largely embedded in shared population dynamics across PMd and M1, while private activity may facilitate flexible, context-dependent processing—such as adaptation to perturbations—without interfering with ongoing execution.

**Ablation study:** To evaluate the contribution of the individual loss functions to the CTAE model we perform an ablation study (full details in Appendix F) and report the prediction accuracy in Tab. 1. We demonstrate that removing any of the three loss functions results in decreased performance.

## 5.2 MULTISENSORY CIRCUIT: SC–ALM

We evaluated CTAE on a new dataset of simultaneous recordings from the superior colliculus (SC) and anterolateral motor cortex (ALM) in mice trained on a multisensory discrimination task (see

| Model | Shared | Private M1 | Private PMd |
|---|---|---|---|
| CTAE | 0.69 (0.03) | 0.22 (0.02) | 0.21 (0.03) |
| CTAE without alignment loss | 0.61 (0.02) | 0.16 (0.02) | 0.2 (0.02) |
| CTAE without orthogonality loss | 0.31 (0.02) | 0.28 (0.02) | 0.29 (0.02) |
| CTAE without shared only reconstruction loss | 0.34 (0.01) | 0.36 (0.01) | 0.37 (0.01) |

Table 1: Ablation study of CTAE loss functions. Each entry indicates the model's accuracy in predicting the reach direction. The values in the brackets indicate standard deviation across 5-folds.

Appendix E.2 for task details). Using high-density Neuropixels probes, neural activity was recorded from superficial and deep layers of SC together with ALM while animals integrated visual and tactile stimuli to identify the target side (left vs. right). We fit our multi-region CTAE ($R > 2$; Appendix B) to this three-region dataset (superficial SC, deep SC, ALM). This multi-region analysis infers a wide range of multi-region interactions: shared-across-all, pairwise-shared, and region-private (Fig. 5).

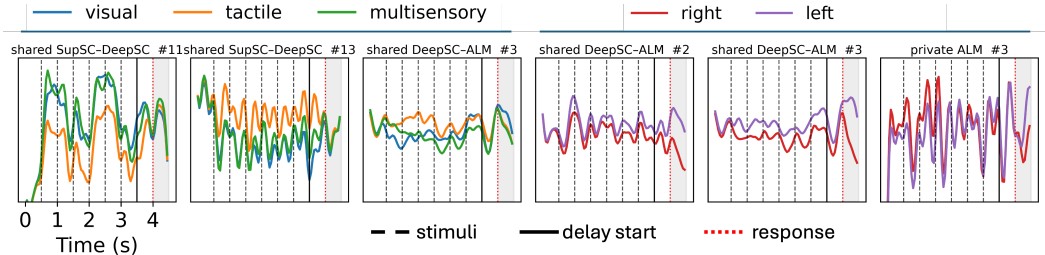

Figure 5: SC-ALM dataset. Representative latents. Each panel shows the condition-averaged time course of one latent (title = subspace and latent index). Left panels: stimuli—visual (blue), tactile (orange), multisensory (green). Right panels: target side—right (red), left (purple).

SC and ALM are key components of a cortico-subcortical loop implicated in linking sensory inputs to motor planning. SC integrates visual and tactile information and contributes to orienting behavior (Stein & Meredith, 1993; Cang & Feldheim, 2013), whereas ALM encodes preparatory and choice-related activity during decision-making tasks (Guo et al., 2014; Li et al., 2015). We hypothesized that shared subspaces across superficial and deep SC layers and between deep layers of SC and ALM would preferentially capture task-relevant features, such as stimulus type and target side, thereby mediating the transformation from multisensory evidence accumulation to decision making.

Figure 6 shows decoding performance of stimuli and target side across private and shared subspaces. For the raw neural activity, Deep SC best discriminates both stimulus and target. Superficial SC activity correctly distinguishes target side but poorly distinguishes between stimulus type for visual and multi-sensory trials, and poorly distinguishes between target sides for tactile trials. In contrast, ALM activity correctly distinguishes target side but poorly distinguishes between stimulus type for tactile and multi-sensory trials, and poorly distinguishes between target sides for visual trials. However, with CTAE we identify shared dynamics between regions that better identify stimulus and target. The shared subspaces between superficial and deep SC, and between deep SC and ALM, yielded higher decoding accuracy than region-private latents. The shared subspace between superficial SC and ALM has little shared dynamics meaningful for discrimination. Our results indicate that deep SC plays a more central role in this circuit, with shared dynamics reflecting evidence accumulation with superficial SC and choice with ALM (Fig. 18). For comparison, we applied mDLAG to the same dataset (Fig. 11). In contrast to CTAE, mDLAG shared and private latents do not exhibit any improvement over predicting from raw activity, and decoding performance remains diffuse across subspaces. This likely reflects the linear observation assumption of mDLAG, which may limit its ability to capture nonlinear multi-region interactions present in this task. Overall, these findings highlight a role for inter-regional subspaces in integrating sensory inputs within SC and propagating them to ALM to support decision-related processes.

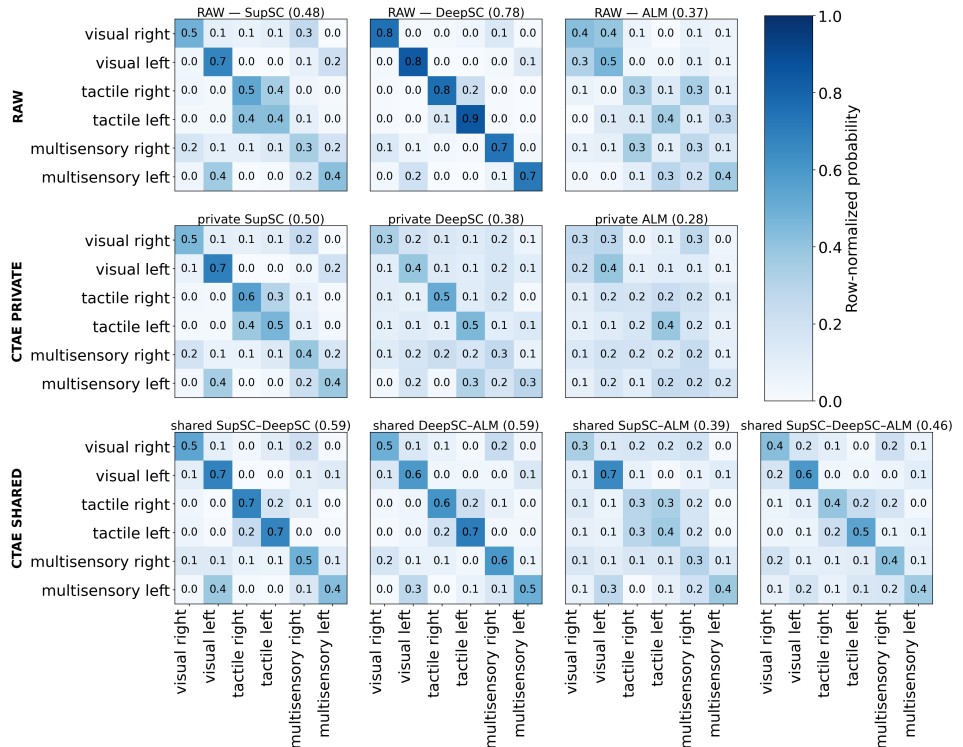

Figure 6: Row-normalized confusion matrices for decoding stimulus × target side. (top) Raw activity from SupSC, DeepSC, ALM; (middle) CTAE private: region-specific latents; (bottom) CTAE shared latents (pairwise and 3-way). Panel titles show mean 5-fold accuracy.

## 6 CONCLUSIONS

In this paper, we presented a new approach for recovering shared and private latents from multi-region neural recordings. Our framework relies on a Transformer-based autoencoder architecture for nonlinear modeling of the relationship between neural activity and latent dynamics, and loss functions designed to recover orthogonal private and shared latents that maximally capture the information in the shared activity between the brain regions. By explicitly separating shared and private sources of variability, CTAE facilitates more robust comparisons across contexts and offers a foundation for addressing deeper questions about how distributed neural populations collectively drive flexible behavior and adapt through experience (Perich et al., 2018). This work demonstrates effective joint latent inference across cortical areas and establishes a foundation for future studies on generalizable, population-level neural computations across behavioral contexts.

There are multiple exciting future directions building on our proposed CTAE model. First, while our paper focuses on neural recordings, our coupled-transformer autoencoder design is not specific to neural data and can be applied to general multiview time series data. Particularly, it can be applied to analyze neural and behavioral recordings to identify features shared between neural activity and behavior, and neural features that are not reflected in behavior and vice versa. Second, currently our approach is scalable to more than two regions by generalizing the orthogonality loss to pairs of brain regions. By making the disentanglement weight vector *learnable*, the dimensions of each of the subspaces can be identified from the training data. Finally, training CTAE directly on spikes with a Poisson/dispersion-aware observation model, i.e. incorporating a likelihood loss matched to spike generation (e.g., Poisson or negative-binomial) will enable directly analyzing unsmoothed spike counts.

## 7 ACKNOWLEDGMENTS

This research is partially supported by a Simons Foundation Pilot Extension Award - 00003245 (G.M. and R.S.), National Science Foundation EFRI 2223822 (G.M., V.G., S.M.N. and J.H.) and Deutsche Forschungsgemeinschaft (DFG, German Research Foundation) as part of the SPP2205 – 533396241 (A.D. and S.M.). This work utilized computational resources provided by the National Research Platform (NRP) Nautilus, supported by the National Science Foundation.

## 8 CONFLICT OF INTEREST

V.G. holds shares in Neuralink Corp. and is Chief Scientific Officer and an options holder at Paradromics, Inc.

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

# A  CTAE ALGORITHM OUTLINE FOR $R = 2$ BRAIN REGIONS

---

**Algorithm 1:** Training Coupled Transformer Autoencoders (CTAE) for two regions

---

**Input:** Paired recordings $\mathbf{X}^{(1)}, \mathbf{X}^{(2)}$, encoder–decoder parameters $\theta, \phi$, latent dimensions $(d_s, d_1, d_2)$, loss weights $\lambda_{\text{shared}}, \lambda_{\text{align}}, \lambda_{\text{orth}}$, optimiser $\mathcal{O}$ and learning rate $\eta$

**Output:** Trained parameters $\theta^\star, \phi^\star$

---

1   Compute weights: $\mathbf{w}_1 \leftarrow \left[\mathbf{1}_{d_s}, \mathbf{1}_{d_1}, \mathbf{0}_{d_2}\right]^\top, \quad \mathbf{w}_2 \leftarrow \left[\mathbf{1}_{d_s}, \mathbf{0}_{d_1}, \mathbf{1}_{d_2}\right]^\top$

2   $\mathbf{w}^{(s)} \leftarrow \mathbf{w}_1 \odot \mathbf{w}_2$

3   **while** *not converged* **do**

4      Encoder forward pass: $\mathbf{Z}^{(1)} \leftarrow E_\theta^{(1)}(\mathbf{X}^{(1)}), \quad \mathbf{Z}^{(2)} \leftarrow E_\theta^{(2)}(\mathbf{X}^{(2)})$

5      Fuse latents: $\mathbf{Z} \leftarrow$ (Eq. equation 4)

6      Extract $\widehat{\boldsymbol{S}}, \widehat{\boldsymbol{P}}^{(1)}, \widehat{\boldsymbol{P}}^{(2)}$ (Eq. equation 5)

7      Data reconstruction: $\widehat{\mathbf{X}}^{(1)} \leftarrow D_\phi^{(1)}\left((\mathbf{w}_1 \mathbf{1}_T^\top) \odot \mathbf{Z}\right), \quad \widehat{\mathbf{X}}^{(2)} \leftarrow D_\phi^{(2)}\left((\mathbf{w}_2 \mathbf{1}_T^\top) \odot \mathbf{Z}\right)$

8      Compute losses: $\mathcal{L} \leftarrow \mathcal{L}_{\text{rec}} + \lambda_{\text{shared}} \mathcal{L}_{\text{shared}} + \lambda_{\text{align}} \mathcal{L}_{\text{align}} + \lambda_{\text{orth}} \mathcal{L}_{\text{orth}}$
     (Eq. equation 8-equation 11)

9      Parameter update: $\theta, \phi \leftarrow \mathcal{O}(\theta, \phi, \nabla_{\theta, \phi} \mathcal{L}_{2R}, \eta)$

10   **return** $\theta^\star, \phi^\star$

---

# B  EXTENDING CTAE TO $R > 2$ BRAIN REGIONS

In the main text, we presented CTAE with a two-region formulation and reported results on datasets involving either two or three regions. Here, we provide the generalization of our framework to $R \geq 3$ regions.

## B.1  PROBLEM FORMULATION (3 REGIONS)

For clarity, we present the formulation below for three regions. However, this construction naturally extends to more than three regions.

Let $\boldsymbol{X}^{(1)} \in \mathbb{R}^{N_1 \times T}, \boldsymbol{X}^{(2)} \in \mathbb{R}^{N_2 \times T}, \boldsymbol{X}^{(3)} \in \mathbb{R}^{N_3 \times T}$ denote simultaneous neural population recordings from three brain regions, each acquired over $T$ time steps with $N_1, N_2, N_3$ channels, respectively. We assume that the observed activity in each region arises from latent dynamics that decompose into:

1. **Region-private dynamics** $\boldsymbol{P}^{(r)}$ that capture computations unique to each region $r \in \{1, 2, 3\}$, lying in subspaces orthogonal to shared components.

2. **Pairwise shared dynamics** $\boldsymbol{S}^{(ij)}$ that span communication subspaces shared between each pair of regions $(i, j) \in \{(1, 2), (2, 3), (1, 3)\}$, reflecting coordinated activity patterns specific to that pair.

3. **Globally shared dynamics** $\boldsymbol{S}^{(123)}$ that capture population trajectories common to all three regions, representing fully shared circuit-level computations.

Formally, the neural activity at each time step $t$ is modeled as nonlinear functions of these latent processes:

$$\mathbf{X}_t^{(1)} = f\left(\boldsymbol{S}_{1:t}^{(12)}, \boldsymbol{S}_{1:t}^{(13)}, \boldsymbol{S}_{1:t}^{(123)}, \boldsymbol{P}_{1:t}^{(1)}\right),$$

$$\mathbf{X}_t^{(2)} = g\left(\boldsymbol{S}_{1:t}^{(12)}, \boldsymbol{S}_{1:t}^{(23)}, \boldsymbol{S}_{1:t}^{(123)}, \boldsymbol{P}_{1:t}^{(2)}\right), \tag{13}$$

$$\mathbf{X}_t^{(3)} = h\left(\boldsymbol{S}_{1:t}^{(13)}, \boldsymbol{S}_{1:t}^{(23)}, \boldsymbol{S}_{1:t}^{(123)}, \boldsymbol{P}_{1:t}^{(3)}\right),$$

where $\boldsymbol{S}_t^{(ij)} \in \mathbb{R}^{d_{ij}}$ denotes the latent state shared between regions $i$ and $j$, $\boldsymbol{S}_t^{(123)} \in \mathbb{R}^{d_{123}}$ the fully shared latent state across all three regions, and $\boldsymbol{P}_t^{(r)} \in \mathbb{R}^{d_r}$ the region-private latent state of region $r$.

Our objective is to recover these latent representations $\{\boldsymbol{S}^{(ij)}, \boldsymbol{S}^{(123)}, \boldsymbol{P}^{(r)}\}$ from the observed neural activity $\{\mathbf{X}^{(1)}, \mathbf{X}^{(2)}, \mathbf{X}^{(3)}\}$, such that the model disentangles private, pairwise shared, and fully shared dynamics in a principled way. For more than three regions, this formulation extends by including shared latent variables for all possible combinatorial subsets of regions, with the training algorithm remaining unchanged. The training algorithm introduced below applies to this general multi-region setting.

## B.2 GENERAL MULTI-REGION SETTING

To move beyond two regions, we extend the formulation with a general mechanism that assigns latent dimensions to all possible combinatorial subsets of regions. This extension builds directly on the same architectural backbone introduced in Section 4, but augments it with an enhanced weighted latent fusion mechanism. This mechanism enables CTAE to disentangle latents shared by arbitrary subsets of regions while preserving strictly region-private structure. All other components—per-region causal Transformer encoders/decoders and the optimization procedure—remain unchanged. The key modifications are described below.

**Encoder outputs.** For each region $r \in \{1, \ldots, R\}$ we keep a dedicated encoder–decoder pair $\left(E_\theta^{(r)}, D_\theta^{(r)}\right)$ and obtain

$$\mathbf{Z}^{(r)} = E_\theta^{(r)}(\mathbf{X}^{(r)}) \in \mathbb{R}^{D \times T}.$$

**Region-specific weight masks.** A single binary matrix encodes which latent dimensions are associated with each region *uses*:

$$\mathbf{W} = \begin{bmatrix} \mathbf{w}_1 \ \mathbf{w}_2 \ \ldots \ \mathbf{w}_R \end{bmatrix}^\top \in \{0, 1\}^{R \times D}, \qquad \mathbf{W}[r, d] = 1 \iff \text{region } r \text{ claims dimension } d.$$

A dimension is *private* when exactly one row in $\mathbf{W}$ is active and *shared* when two or more rows are active, thereby accommodating every subset pattern (for instance, when $R = 3$ the binary code 110 indicates a dimension used by regions 1 and 2 but not 3; 101 corresponds to regions 1 and 3; and 111 denotes a dimension shared by all three). Relative to the two–region mask 3, $\mathbf{W}$ expands the notion of "shared" from "both regions" to "arbitrary subset of regions". The membership matrix $\mathbf{W}$ can be (i) fixed a priori from anatomical knowledge, (ii) selected by a hyper-parameter search that allocates an appropriate number of latent dimensions to each subset pattern (analogous to the two-region masks in Eq. equation 3), or (iii) treated as a fully learnable variable and optimised jointly with the rest of the network.

**Weighted latent fusion.** We fuse the region–specific latents with a masked average

$$\mathbf{Z}_t[d] = \frac{\sum_{r=1}^{R} \mathbf{W}[r, d] \, \mathbf{Z}_t^{(r)}[d]}{\sum_{r=1}^{R} \mathbf{W}[r, d]}, \tag{14}$$

so that private dimensions stay unchanged, whereas any dimension claimed by multiple regions is *forced* to align across them.

**Region-specific decoding.** Each decoder receives only the dimensions it owns,

$$\widehat{\mathbf{X}}^{(r)} = D_\phi^{(r)}\left((\mathbf{w}_r \mathbf{1}_T^\top) \odot \mathbf{Z}\right),$$

exactly as in the two–region case. Hence the extension does not change model capacity but merely broadens how "shared" information is defined.

This design (i) embeds all regions in one latent space of size $D$, (ii) preserves region-specific structure through fixed relevance masks, and (iii) enforces cross-region consistency via Eq. equation 14.

## B.3 TRAINING OBJECTIVE

We retain the four losses used for two regions—reconstruction, shared-only reconstruction, alignment, and orthogonality—while replacing two binary masks with the matrix $\mathbf{W}$.

**Reconstruction.** The reconstruction loss is unchanged, only we now sum over all regions.

$$\mathcal{L}_{\text{rec}} = \sum_{r=1}^{R} \left\| \widehat{\mathbf{X}}^{(r)} - \mathbf{X}^{(r)} \right\|_F^2. \tag{15}$$

**Shared-only reconstruction.** To guarantee that *all* cross-region regularities live inside shared dimensions, we mask out every coordinate used by fewer than two regions. Let $\mathbf{s} = \mathbf{1}_{\{\sum_r \mathbf{w}_{r,\cdot} \geq 2\}} \in \{0,1\}^D$ and $\mathbf{w}_r^{(s)} = \mathbf{w}_r \odot \mathbf{s}$. We then reconstruct each region from the shared subspace alone:

$$\mathcal{L}_{\text{shared}} = \sum_{r=1}^{R} \big\| D_\phi^{(r)}\big((\mathbf{w}_r^{(s)}\mathbf{1}_T^\top) \odot \mathbf{Z}\big) - \mathbf{X}^{(r)}\big\|_F^2. \tag{16}$$

When $R = 2$ this term reduces to Eq. (11) in the main text, but for $R > 2$ it now supervises *every* shared subset simultaneously.

**Alignment.** Each region's encoder estimate should coincide with the fused latent over the dimensions it owns:

$$\mathcal{L}_{\text{align}} = \sum_{r=1}^{R} \big\|(\mathbf{w}_r\mathbf{1}_T^\top) \odot \mathbf{Z} - (\mathbf{w}_r\mathbf{1}_T^\top) \odot \mathbf{Z}^{(r)}\big\|_F^2. \tag{17}$$

Private coordinates contribute zero because they match by definition; shared coordinates are actively driven together, extending the simple two-region difference penalty to an $R$-way match.

**Orthogonality.**

$$\mathcal{L}_{\text{orth}} = \big\|\tfrac{1}{T}\mathbf{Z}\mathbf{Z}^\top - \text{diag}\big(\tfrac{1}{T}\mathbf{Z}\mathbf{Z}^\top\big)\big\|_F^2. \tag{18}$$

Identical to the two-region version, this term keeps all latent dimensions—private or shared—mutually decorrelated.

**Total loss.**
$$\mathcal{L} = \mathcal{L}_{\text{rec}} + \lambda_{\text{shared}}\mathcal{L}_{\text{shared}} + \lambda_{\text{align}}\mathcal{L}_{\text{align}} + \lambda_{\text{orth}}\mathcal{L}_{\text{orth}}, \tag{19}$$

with $\lambda_{\text{shared}}, \lambda_{\text{align}}, \lambda_{\text{orth}}$ selected on a validation set. The objective reduces exactly to the two-region formulation when $R = 2$; for $R > 2$ it differs only in how masks are constructed and applied, maintaining conceptual continuity while capturing richer patterns of shared neural dynamics.

## C  SYNTHETIC NONLINEAR COUPLED DATASET

We construct a synthetic two-region dataset in which the ground-truth shared and private latent trajectories are known. We generate paired sequences $X^{(1)} \in \mathbb{R}^{N \times T \times D_1}$ and $X^{(2)} \in \mathbb{R}^{N \times T \times D_2}$ from latent trajectories $S \in \mathbb{R}^{N \times T \times d_s}$, $P^{(1)} \in \mathbb{R}^{N \times T \times d_1}$, and $P^{(2)} \in \mathbb{R}^{N \times T \times d_2}$. Unless otherwise stated, we use $N = 1000,\quad T = 32,\quad d_s = 4,\quad d_1 = d_2 = 3,\quad D_1 = D_2 = 32$.

The latent trajectories $S_t$, $P_t^{(1)}$, and $P_t^{(2)}$ are generated as sinusoidal processes with randomly sampled amplitudes, phases, and integer frequencies. Concretely, for each trial $i \in \{1, \dots, N\}$ and latent dimension $k$, we sample an amplitude $a_{i,k} \sim \mathcal{N}(0,1)$, a phase $\phi_{i,k} \sim \text{Uniform}(0, 2\pi)$, and a frequency $f_{i,k} \sim \text{Uniform}\{2,3,4\}$, and define

$$z_{i,t,k} = a_{i,k}\sin\big(2\pi f_{i,k}\tau_t + \phi_{i,k}\big), \qquad \tau_t = \frac{t}{T},\ \ t = 1, \dots, T, \tag{20}$$

The shared latents $S$ and private dimensions $P^{(1)}$ and $P^{(2)}$ are generated independently using the above construction with dimensionality $d_s$, $d_1$ and $d_2$, respectively. Each region linearly mixes shared and private latents:

$$H_t^{(1)} = \alpha_s A_s^{(1)} S_t + \alpha_p A_p^{(1)} P_t^{(1)}, \qquad H_t^{(2)} = \alpha_s A_s^{(2)} S_t + \alpha_p A_p^{(2)} P_t^{(2)},$$

where $A_s^{(r)} \in \mathbb{R}^{D_r \times d_s}$ and $A_p^{(r)} \in \mathbb{R}^{D_r \times d_r}$ have i.i.d. Gaussian entries scaled by $1/\sqrt{d}$. We set $\alpha_s = \alpha_p = 2.0$.

We then introduce nonlinear distortion by applying an elementwise mapping:

$$X_{t,\text{clean}}^{(r)} = \tanh\big(H_t^{(r)}\big) + 0.1\big(H_t^{(r)}\big)^2$$

The resulting $X_{t,\text{clean}}^{(r)}$ can be interpreted as synthetic *log-firing rates*. We then add independent Gaussian noise:

$$X_t^{(r)} = X_{t,\text{clean}}^{(r)} + \varepsilon_t^{(r)}, \qquad \varepsilon_t^{(r)} \sim \mathcal{N}(0, \sigma^2 I),$$

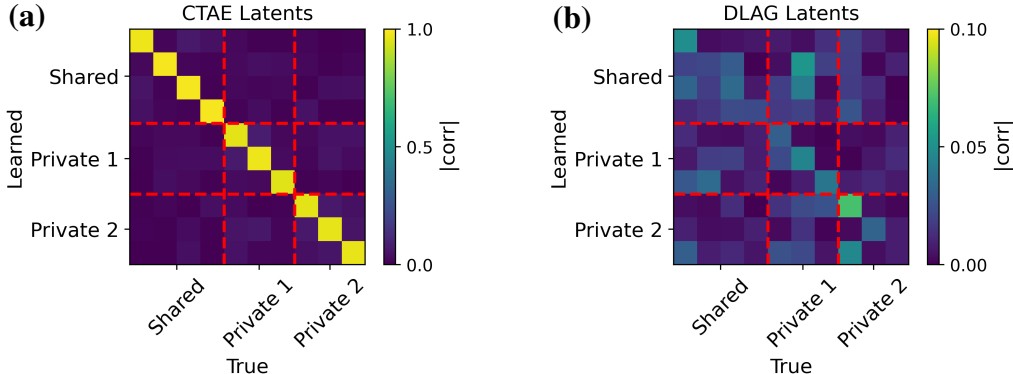

Figure 7: Absolute Pearson correlation between aligned learned latents and ground-truth latents on the synthetic dataset. (a) CTAE. (b) DLAG. Red dashed lines indicate shared and region-specific partitions.

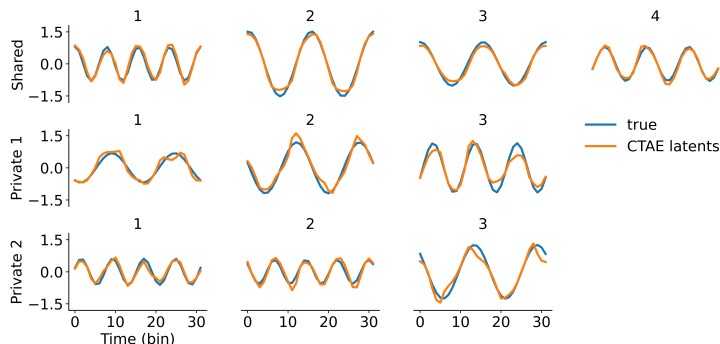

Figure 8: Latent trajectories on the synthetic dataset. Rows correspond to shared, private region 1, and private region 2 latents. Blue: ground truth. Orange: aligned CTAE latents.

with $\sigma = 0.01$.

After training, we extract the learned latent trajectories from each model and align each latent subspace (shared and private) to the corresponding ground-truth latents using a linear transformation estimated by least-squares regression on a held-out subset of trials. This alignment accounts for the fact that latent representations are only defined up to a linear transformation.

We then compute the absolute Pearson correlation between each aligned learned latent dimension and each true latent dimension. Figure 7 shows the resulting correlation matrices. For CTAE, the correlations exhibit a clear block structure: dimensions assigned to the shared subspace align with the true shared latents, and region-specific dimensions align with their corresponding private latents, with minimal or no cross-block correlation. In contrast, DLAG does not produce a structured alignment; correlations are weak and do not follow the shared/private partition. DLAG assumes a linear observation model, whereas in this dataset the observations are nonlinear functions of the latent variables. This model mismatch limits its ability to recover the underlying shared and private structure. Figure 8 further illustrates representative latent trajectories after alignment. CTAE closely matches the ground-truth shared and private dynamics across time, while preserving separation between shared and region-specific components. Overall, CTAE recovers latent dimensions that closely match the true shared and private dynamics under nonlinear observations, whereas DLAG fails to recover the underlying structure in this setting.

## D   TRAINING AND HYPERPARAMETER DETAILS

To ensure consistency in model capacity across regions, we adopt identical causal Transformer-based encoder–decoder architectures for each region, using an equal number of encoder and decoder layers to maintain architectural symmetry. We then performed a grid search over:

**Number of layers.** We varied the total number of Transformer layers $L \in \{1, 2, 3\}$ in both encoder and decoder to trade off model capacity against overfitting risk and computational cost.

**Latent dimensions.** Private and shared dimensions, $d \in \{5, 10, 15\}$. Note - Here, the specified latent dimensionality $d$ should be treated as an upper bound on bottleneck capacity. After training with the designed regularizers, redundant factors collapse to near-zero explained variance, yielding an effective dimensionality $d_{\text{eff}} \leq d$ that approximates the data's intrinsic dimensionality.

**Loss weights.**

- Shared-space: $\lambda_{\text{shared}} \in \{0, 1, 2\}$
- Alignment: $\lambda_{\text{align}} \in \{0, 0.05, 0.1, 0.5\}$
- Orthogonality: $\lambda_{\text{orth}} \in \{0, 0.001, 0.01, 0.05\}$

**Learning rate.** $\eta \in \{10^{-3}, 10^{-4}\}$

**Warm-up schedule for orthogonality loss.** To prevent an overly strict orthogonality constraint from hindering early representation learning, we set $\lambda_{\text{orth}} = 0$ for the first $e$ epochs—allowing the model to focus on reconstruction and shared/private separation—then linearly ramp it up to its target $\lambda_{\text{orth}}$ value over the next $e$ epochs. This gradual increase stabilizes training by delaying the full orthogonality penalty until the autoencoders have already learned meaningful features. We explored $e \in \{100, 500, 1000\}$. Formally,

$$\lambda_{\text{orth}}^{(t)} = \begin{cases} 0, & t \leq e, \\ \dfrac{t-e}{e}\,\lambda_{\text{orth}}, & e < t \leq 2e, \\ \lambda_{\text{orth}}, & t > 2e\,. \end{cases}$$

**Positional encoding.** We use fixed, deterministic sinusoidal signals into each input embedding to convey temporal order, following (Vaswani et al., 2017). Specifically, for time step $t$ and embedding dimension index $i$ (with model dimension $d_{\text{model}}$):

$$\text{PE}(t, 2i) = \sin\!\Big(\tfrac{t}{10000^{2i/d_{\text{model}}}}\Big), \quad \text{PE}(t, 2i+1) = \cos\!\Big(\tfrac{t}{10000^{2i/d_{\text{model}}}}\Big).$$

These encodings provide both absolute and relative position information without adding any learnable parameters, enabling the self-attention layers to distinguish different time steps.

All models were trained for $10\,000$ epochs, and the run minimizing the total loss in equation 12 on a held-out validation set was selected. Table 2 summarizes all the hyperparameters and their corresponding search ranges and Table 3 reports the optimal settings for the two real and one synthetic datasets studied. For the corresponding DLAG and mDLAG baselines, we matched the total number of latent dimensions to those used by CTAE for each dataset (Table 3) to ensure a fair comparison.

Table 2: Hyperparameter search ranges.

| Hyperparameter | Values |
|---|---|
| # Transformer layers $L$ | $\{1, 2, 3\}$ |
| latent dims. $d_1 = d_2 = d_s$ | $\{5, 10, 15\}$ |
| $\lambda_{\text{shared}}$ | $\{0, 1, 2, 5\}$ |
| $\lambda_{\text{align}}$ | $\{0, 0.05, 0.1, 0.5\}$ |
| $\lambda_{\text{orth}}$ | $\{0, 0.001, 0.01, 0.05\}$ |
| Learning rate $\eta$ | $\{10^{-3}, 10^{-4}\}$ |
| Warm-up epochs for orth. loss | $\{100, 500, 1000\}$ |

**Compute Resources.** All models were trained using CUDA-accelerated implementations in PyTorch. Training was performed on NVIDIA Quadro RTX 8000 GPUs, supplemented by additional compute resources from a National Research Platform (NRP) Nautilus.

Table 3: Selected hyperparameters for M1–PMd, SC-ALM and Synthetic datasets.

| Parameter | M1–PMd | SC–ALM | Synthetic |
|---|---|---|---|
| latent dims | 10 | 15 | $d_s = 4, d_1 = d_2 = 3$ |
| # Layers $L$ | 3 | 1 | 1 |
| $\lambda_{\mathrm{shared}}$ | 1 | 1 | 5 |
| $\lambda_{\mathrm{align}}$ | 0.5 | 0.1 | 0.1 |
| $\lambda_{\mathrm{orth}}$ | 0.01 | 0.01 | 0.01 |
| Learning rate $\eta$ | $10^{-4}$ | $10^{-4}$ | $10^{-3}$ |
| Warm-up (orth. loss) | 100 epochs | 1000 epochs | 100 epochs |

## E  DATASET DETAILS

The motor circuit dataset used in this study is publicly available and was originally released with (Perich et al., 2018). Both datasets were preprocessed using standard neural signal processing pipelines. Specifically, spike trains were first binned at 100 ms for both M1–PMd dataset and SC-ALM dataset. The binned spike counts were then smoothed using a Gaussian filter (kernel size of 3 and 2 for M1-PMd and SC-ALM, respectively) to estimate instantaneous firing rates, which were used as input to the CTAE model.

### E.1  MOTOR CIRCUIT DATASET: CENTER-OUT ARM REACHING TASK

We use the dataset from (Perich et al., 2018), which includes neuronal recordings from a Utah Array in M1 and PMd; see Fig. 9, Left). In this task, a monkey performs center-out arm reaches using a planar manipulandum. Neural activity was recorded from the primary motor cortex (M1) and dorsal premotor cortex (PMd) via Utah arrays, yielding spike-sorted activity from 52 putative units in M1 and 66 in PMd (Fig. 9 Right). Each trial begins with a target onset, followed by a variable delay period (0.5–1.3 s) for movement preparation. After a "go" cue, the monkey initiates a reach to one of eight evenly spaced targets in a 2D workspace, holds briefly, and returns to center. The dataset includes 208 trials across 8 target conditions, with approximately 26 repetitions per condition.

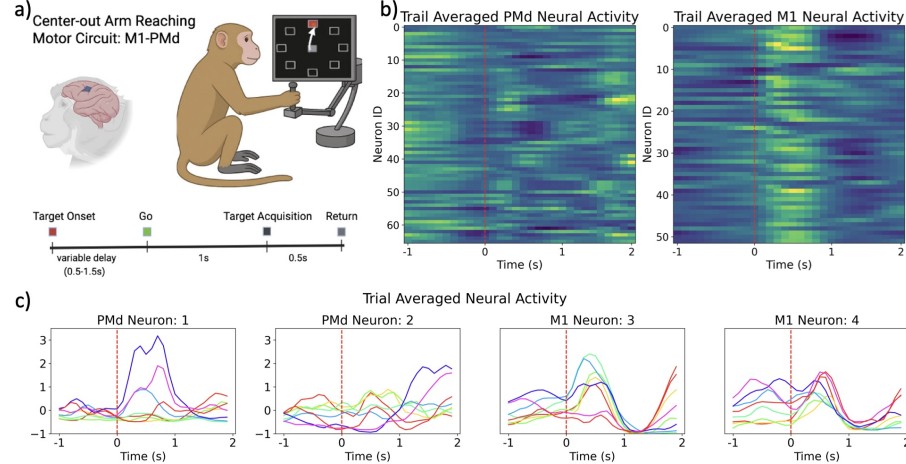

Figure 9: Motor study. a) Schematic of the behavioral tasks performed by non-human primates in the datasets used in Section 5. Center-Out Arm Reach Task: The monkey controls a manipulandum to perform instructed delay reaches toward one of eight peripheral targets, with neural activity recorded from M1 and PMd using Utah arrays. b) Population firing rates in M1 and PMd. Heatmaps shows the trial-averaged z-scored firing rates for individual neurons (rows) across time. c) Condition-averaged firing rates of individual neurons in M1 and PMd. Each panel shows trial-averaged firing rates for single neurons, grouped and colored by reach direction. Colormap follows that in Fig. 4.

### E.2 SC-ALM Dataset: Multisensory Task

The anterolateral motor cortex (ALM) and the superior colliculus (SC) are important parts of a cortico-subcortical loop that transforms multisensory inputs into behavioral decisions. To study the role of these areas in multisensory integration and decision-making, we trained mice in a multisensory discrimination task, where animals had to integrate visual and tactile information over time to identify the target stimulus side. We then performed simultaneous neural recordings in ALM and SC, using high-density Neuropixels probes, in task-performing animals., see Fig. 10(left).

Each trial had a total duration of 7 seconds, comprising a 1-second baseline, then 3 seconds of probabilistic stimuli, followed by a 0.5-second delay without any stimulus and finally, the spouts moved in front of the mouse, allowing up to 2.5 seconds for a licking response. The mice were rewarded when responding on the side where more sensory stimuli were presented during the stimulus period. The stimulus period was composed of six periods of 500 milliseconds each, during which a stimulus could be presented to the mouse. One to six stimuli were presented on a given side, with each stimuli likelihood of presentation determining the difficulty of the task. Depending on the trial type, a stimulus could either be a *visual* grating moving along a screen (visual trial), an *air puff* of 100ms to the vibrissae on one side of the mouse (tactile trial), or both, in a simultaneous congruent *multisensory* manner (multisensory trial). Mice were trained to respond by licking the side on which more stimuli were shown. An example trial is illustrated in Fig. 10(center). An example of a multisensory trial timeline is presented in Fig. 10(right).

We analyzed a single session from an expert animal that achieves $\sim 70\%$ correct choice accuracy (564 trials in total), with approximately 80–100 trials per stimulus type and per target direction

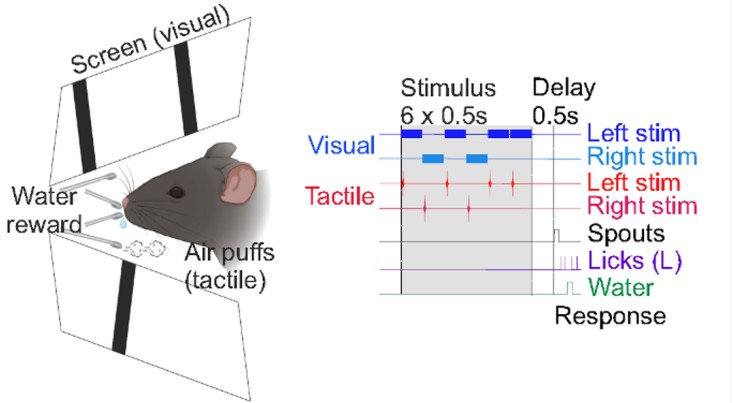

Figure 10: Multisensory study. Left: Schematic of the 2-alternative forced choice task performed by the mice, trained to lick on the side where more stimuli was presented. The mice received a water reward through the target spout. Right: Schematic of the task timeline in a multisensory trial with a left-side target. The stimulus phase is in grey, the delay and response phase are in white, separated by a vertical line and the movement of the spouts begins at the start of the response period.

## F ABLATION STUDY: INDIVIDUAL LOSS CONTRIBUTIONS

We evaluate the contribution of each loss term by training four CTAE variants:

1. **Full CTAE**: all losses active (shared-only reconstruction, alignment, orthogonality, standard reconstruction).
2. **No Shared-only Reconsruction**: drop the loss that reconstructs inputs using only the shared subspace.
3. **No Alignment**: remove the term that aligns shared latents across regions.
4. **No Orthogonality**: remove the orthogonality constraint between shared and private latents.

For each variant, we (i) visualize ablated interactions where meaningful, and (ii) quantify reach-direction decoding accuracy from each latent subspace.

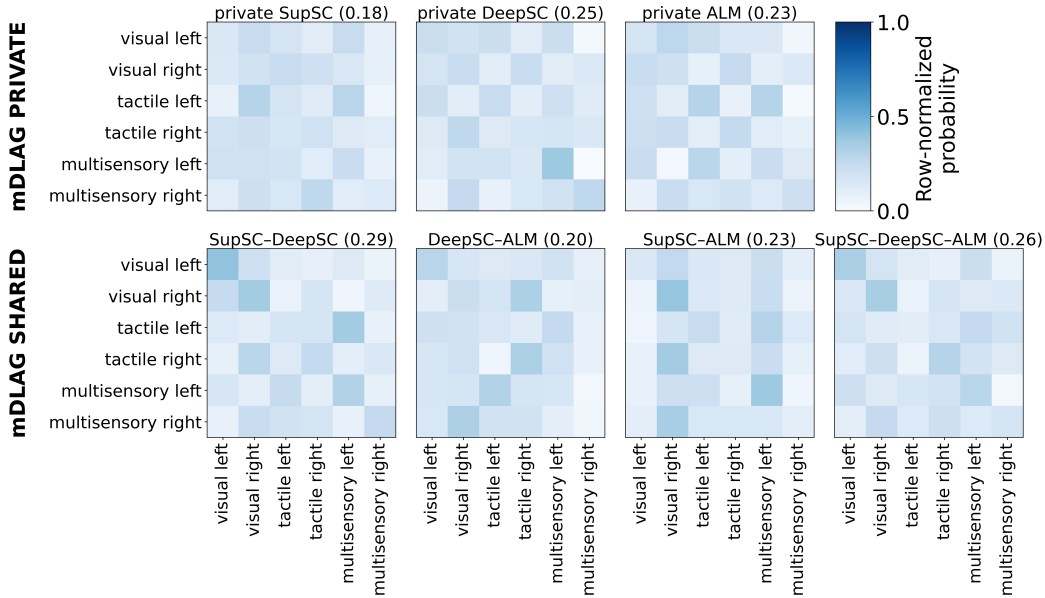

Figure 11: Row-normalized confusion matrices for decoding stimulus × target side. (top) mDLAG private: region-specific latents; (bottom) mDLAG shared latents (pairwise and 3-way). Panel titles show mean 5-fold accuracy.

## F.1 SHARED-ONLY RECONSTRUCTION LOSS ABLATION

The shared-only reconstruction loss ensures that the shared subspace alone can reconstruct the original neural input; it encourages capturing all common dynamics in that subspace. Without it, information shifts into private latents, reducing interpretability of the shared space. As shown in Table 1, removing this loss component leads to a dramatic drop in shared-latent decoding accuracy, accompanied by a corresponding increase in private-latent accuracies.

## F.2 ALIGNMENT LOSS ABLATION

The alignment loss penalizes discrepancies between shared latents from each region, enforcing that they capture the same underlying dynamics. To highlight the significance of this term, we performed loss ablation experiments and analyzed the latents when no alignment loss was enforced. In Fig. 12a), the alignment loss is enforced, forcing the latents to have high similarity and significant overlap. When the alignment loss is relaxed in Fig. 12b), the two shared latents diverge and sometimes cancel each other, indicating a breakdown in cross-region correspondence. Correspondingly, Table 1 shows that removing the alignment loss degrades shared-latent decoding accuracy, while private-latent accuracies remain largely unchanged.

## F.3 ORTHOGONALITY LOSS ABLATION

The orthogonality loss enforces that shared and private latents carry non-redundant information by minimizing their dot products. To evaluate the importance of this loss, ablation experiments with relaxed orthogonality constraints were used during model training. The degree of disentanglement is quantified using the dot products between latents. In Fig. 13 a) the orthogonality loss is enforced in the model and this constrained the latents to be maximally disentangled from each other. In Fig. 13 b) the orthoginality loss is relaxed in the model and there is a significant overlap between the latents, indicating a breakdown in disentanglement. Correspondingly, Table 1 shows that removing the orthogonality loss degrades shared-latent decoding accuracy, while private-latent accuracies remain largely unchanged.

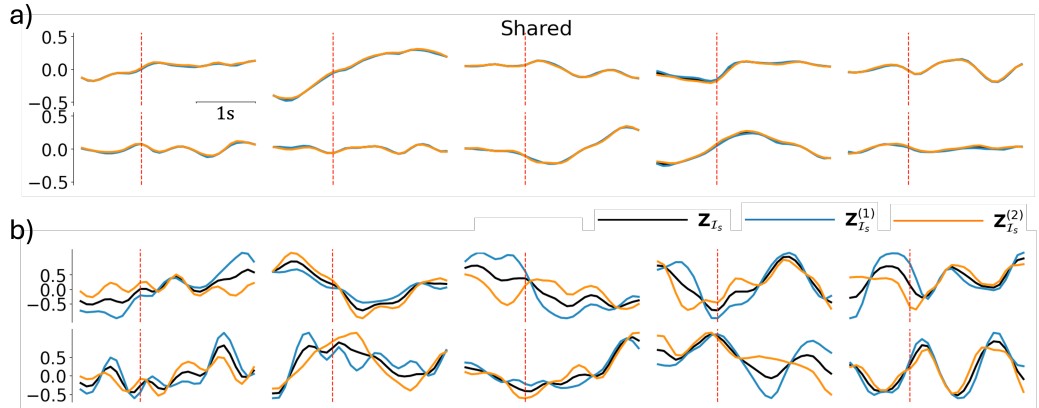

Figure 12: Alignment of shared latents in M1-PMd. Each subplot visualizes a single latent. Blue and orange traces indicate shared latents obtained from M1 and PMd respectively. Black trace represents the mean of blue and orange traces. (a) Alignment loss is enforced in the model. The shared latents have significant overlap. (b) Alignment loss is relaxed in this model. The latents deviate significantly from each other.

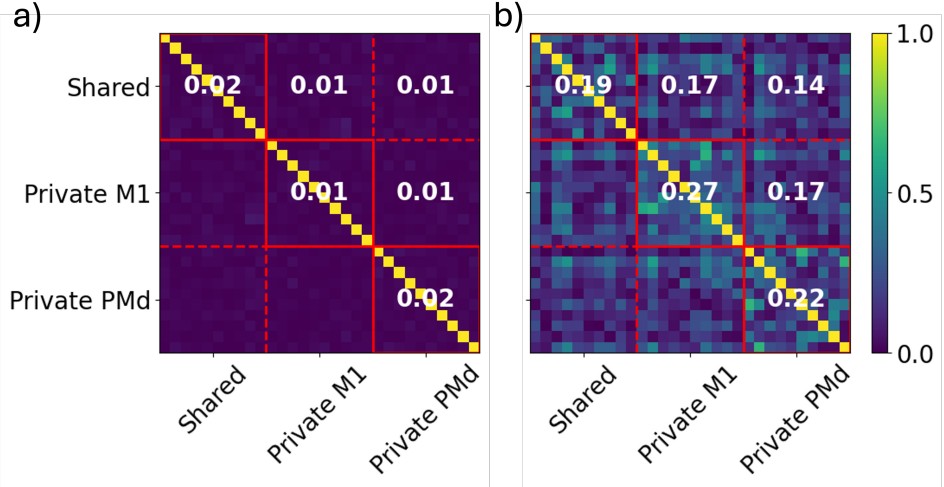

Figure 13: Orthogonality between CTAE latents in M1-PMd. The red lines are used to differentiate between the shared and region-private M1 and region-private PMd latents, in that order. The values in the blocks indicate the mean of dot product value between the latents. (a) Latents from model with orthogonality loss enforced (b) Latents from model with orthogonality loss relaxed

## G ANALYSIS OF LATENTS AND ATTENTION MAPS

### G.1 OVERVIEW OF LATENT DYNAMICS

The CTAE model extracts both shared and region-private latents from neural activity, and does so at a single-trial resolution. Shared and region-private latents for the motor dataset are obtained from the best performing CTAE model and visualized in Fig. 14. The region-private latents capture neural activity patterns that are specific to the region and thereby, reflect local dynamics and computations. The shared latents capture activity patterns that are common to both regions. The representation of task variables such as reach direction in motor task was analyzed by averaging latent trajectories across trials grouped by task variables. In both regions in Fig. 14, distinct latents exhibit systematic variation across the task variables, indicating that the latents are inherently capturing behaviorally relevant information.

Fig. 15 visualizes a subset of the CTAE inferred neural latent dynamics (from the set of all latents in Fig. 14). Each panel shows state-space trajectories for different latent groups across task epochs. Immediately after target onset, trajectories in the shared subspace $Z^{(s)}$ diverge from a common baseline toward target-dependent fixed points, yielding an explicit goal code—consistent with prior reports of a PMd–M1 preparatory subspace. During movement, variance in $Z^{(s)}$ increases along axes orthogonal to the preparatory subspace, indicating a shift in PMd–M1 coupling as the task transitions to execution (Fig. 15B). We also observe that PMd-private latents maintain a stable target representation during the preparatory phase (Fig. 15C), consistent with staging signals that are subsequently reflected in the shared space; after the go cue, this private structure weakens, no longer reflecting clear separation of different targets, consistent with transient reorganization or monitoring as movement unfolds (Fig. 15D).

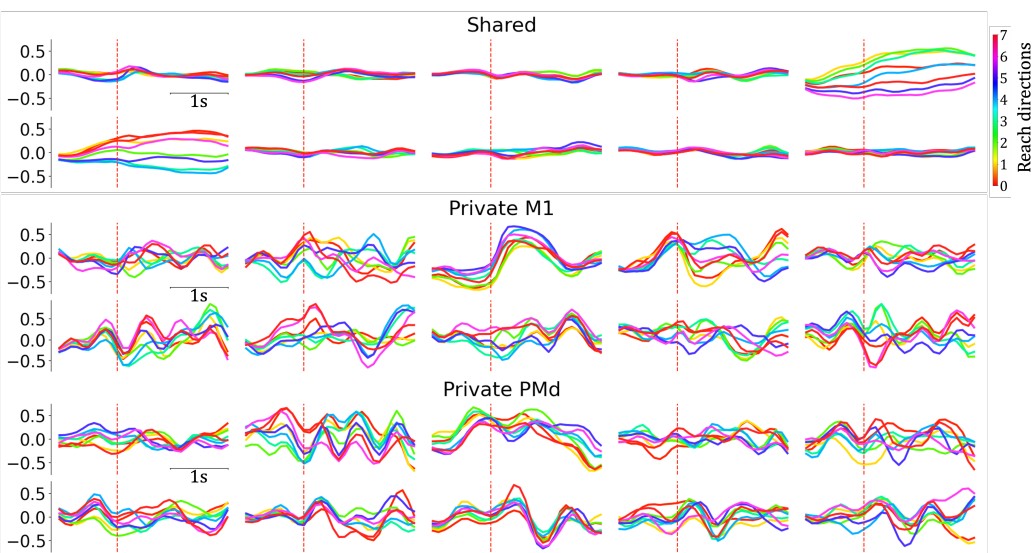

Figure 14: CTAE latent dynamics in M1-PMd. Each subplot visualizes a single neural latent, with colors indicating trial-averaged latents for each reach direction. Red dashed line indicates go-cue. (Top) Shared latents capture neural dynamics present in both M1 and PMd. (Middle) M1-private latents capture dynamics specific to M1 (Bottom) PMd-private latents capture dynamics specific to PMd.

In Fig. 16 and Fig. 17, we present all the latents for the SC-ALM dataset, where we trial-average the latents within target side and stimulus type, respectively. In general, most latents exhibit oscillatory patterns that, to varying degrees, are tuned to the stimulus presentation times; superficial SC latents tend to exhibit higher-frequency oscillations. Shared latents for Deep SC and ALM, and region-specific ALM latents, exhibit separation between the target sides at the delay and response. Superficial SC latents and shared superficial and deep SC latents exhibit separate activity patrern for tactile trials comapred to visual and multisensory trials. In contrast, Deep SC and ALM shared latents exhibit patterns that separate all three stimulus types.

We next evaluate temporal decoding of the target variable (e.g., reach direction, target side) from the latents, to better determine the dynamics represented by the latents in shared and private subspaces.

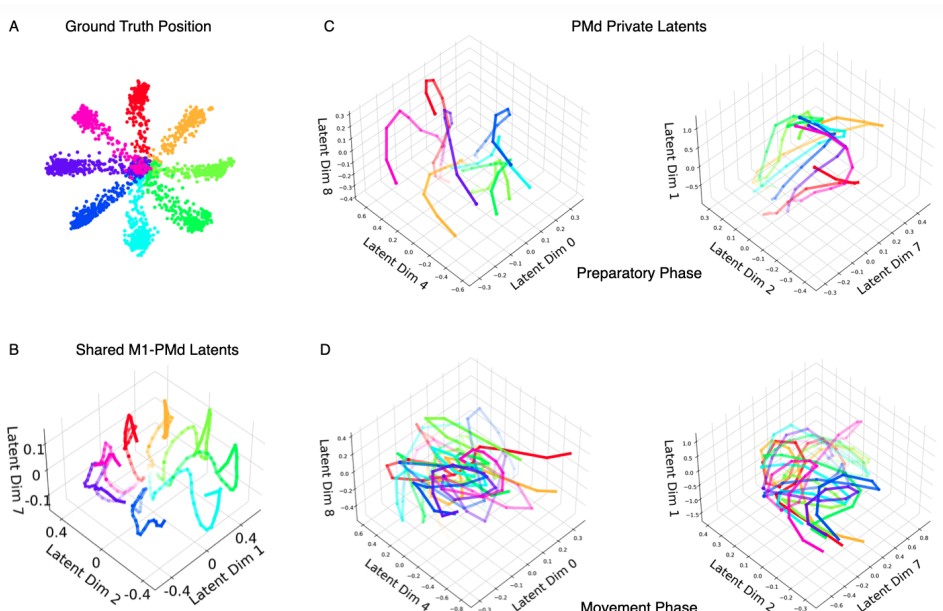

Figure 15: State-space views of CTAE latents across task epochs. Each subplot shows trajectories in a 3 dimensional latent subspace. Colors denote trial-averaged trajectories for each reach direction in (A). (B) Shared PMd–M1 subspace: trajectories evolve over the entire trial, circle denotes 'go' cue. (C) PMd-private subspace during the preparatory phase. Left and right plots are two different 3D subspaces within the private latents. (D) PMd-private subspace during the movement phase. Together, these panels highlight distinct PMd–M1 interaction modes that differ between preparation and execution. Left and right same as in (C).

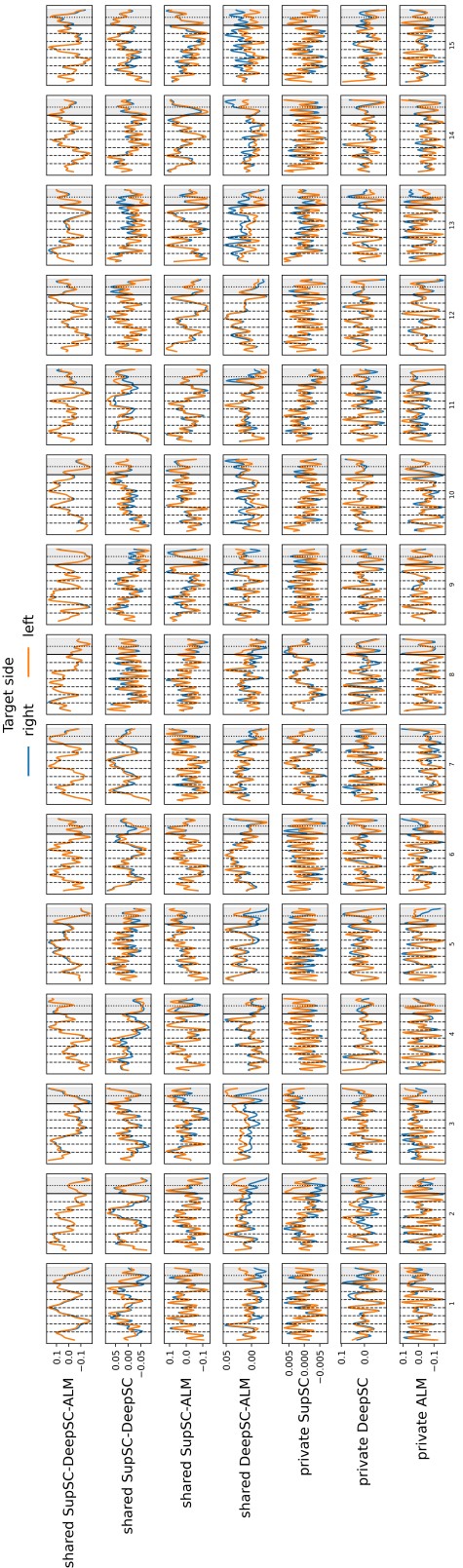

Figure 16: SC-ALM: CTAE latents averaged across target side.

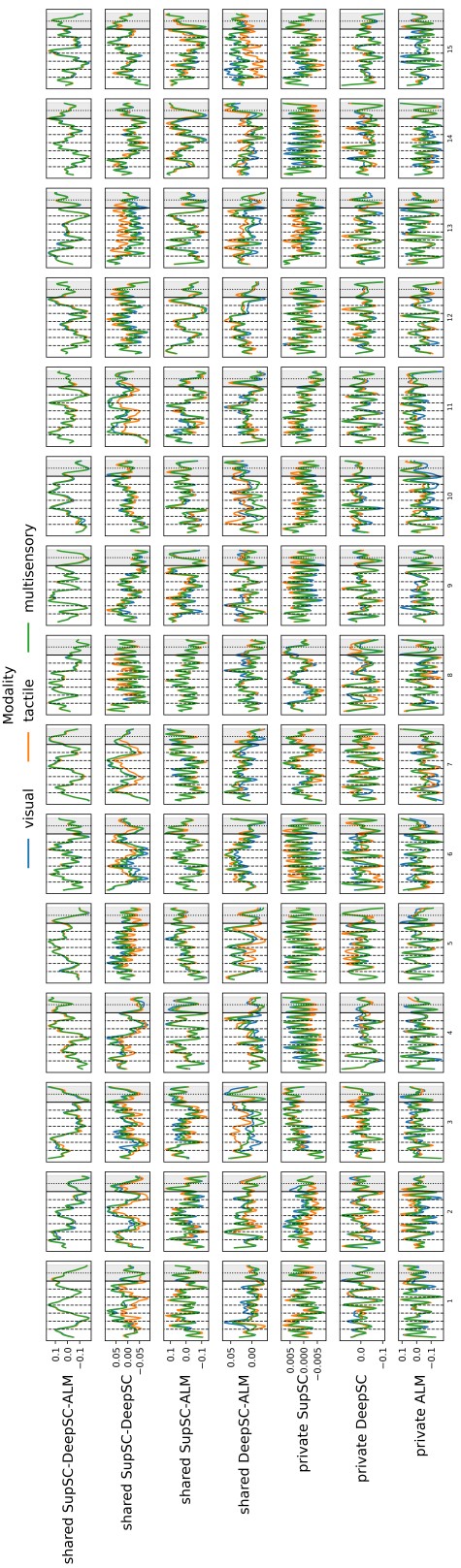

Figure 17: SC-ALM: CTAE latents averaged across stimulus type.

## G.2 TEMPORAL DECODING ANALYSIS FROM CTAE LATENT SUBSPACES

**Time-resolved decoding.** To localize when task information appears in each CTAE latent subspaces, we considered per-trial, per-time bin latents from CTAE's private and shared (pairwise and all-shared) subspaces. At every time bin, the input features are the latent dimensions in that subspace (i.e., private, all shared, or pairwise shared) within a window of 5 time bins. For each subspace we ran 5-fold stratified cross-validation with a single multinomial logistic-regression classifier (with feature standardization) trained once per fold on the full time-flattened feature matrix. Time-resolved accuracy was then obtained without re-training: at test time we fixed the classifier and masked all features outside a temporal context centered at each time point (a symmetric sliding window of five bins, i.e., $\pm 2$ bins). We report mean $\pm$ s.d. across folds at every time bin and overlay event markers for cue onsets, delay start, and delay end.

**Findings: SC-ALM dataset.** Across subspaces, the accuracy of the shared latents progressively improves as the response approaches, indicating that cross-area information integrates toward the decision and movement execution. Specifically, the shared subspace across all 3 regions demonstrates a positive trend towards the response period with the highest accuracy at the response period and a slight increase in accuracy after each stimulus time. Superficial SC private subspace shows the highest prediction accuracy prior to the delay period, consistent with sustained maintenance of task-relevant signals. In contrast, ALM and the Deep SC–ALM shared subspace peak after the response, suggesting strong encoding of the target side and decision (the two are highly correlated in expert mice). The Superficial SC–Deep SC shared subspace also exhibits marked prediction increases immediately following most stimulus onsets, pointing to early multisensory interactions between superficial and deep SC layers, and shared representation of stimulus direction. Together, these patterns reveal complementary temporal roles: private subspaces emphasize delay-period maintenance and response choice, while shared subspaces carry stimulus- and response-proximal information that ramps as action nears.

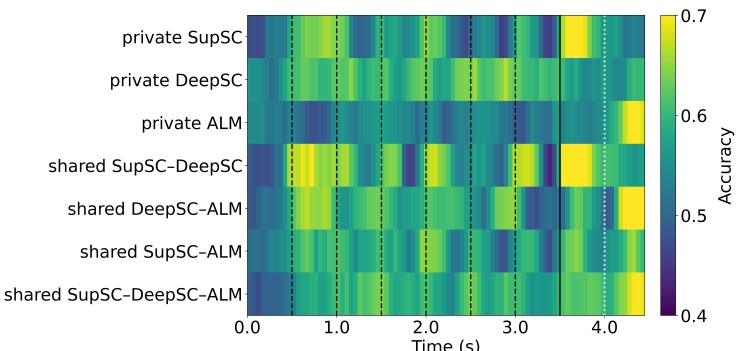

Figure 18: Time-resolved decoding accuracy (mean over 5-fold CV) for each latent subspace. Rows correspond to private and shared subspaces (private SupSC/DeepSC/ALM; shared pairs and all-shared), columns are time. Dashed vertical lines mark cue onsets (every 0.5 s from 0.5–3.0 s), the solid line marks delay start (3.5 s), and the white dotted line marks the start of the decision making period (4.0 s).

**Findings: Motor-Circuit dataset.** Time-resolved reach-direction prediction accuracy shows a pronounced increase following the Go-Cue (Fig. 19). This pattern is observed across latents obtained from CTAE, DLAG and DeepCCA. Shared latents from all three models peak in accuracy following the go-cue, indicating presence of higher directional information, with CTAE-shared consistently outperforming those from the other models. In contrast, PMd-specific latents show an increase in accuracy prior to the go-cue, while, M1-specific latents achieve peak-accuracies post go-cue. These findings suggests that PMd encodes anticipatory directional information, while M1 encodes directional information more strongly during movement. Overall, these results align with previous reports on directional information representation in M1 and PMd (Lara et al., 2018).

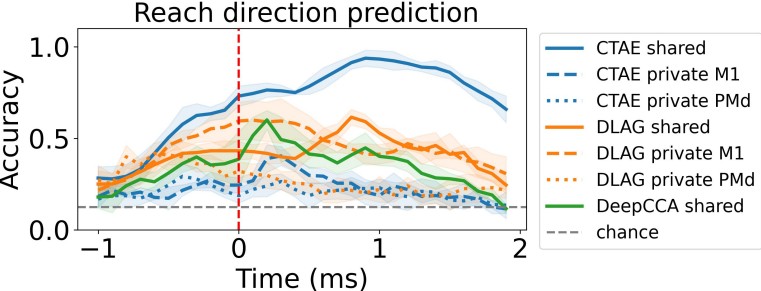

Figure 19: Time-resolved reach-direction decoding accuracy (5-fold CV) for each subspace, across multiple models. Red dashed line represents go-cue. Shared latents from all models show a sharp increase in accuracy following the Go-Cue, with CTAE-shared consistently outperforming others. PMd-specific latents exhibit a pre-Go-Cue rise, indicating anticipatory directional information, whereas M1-specific latents peak post Go-Cue, reflecting direction-specific encoding after movement initiation. Shading represents $\pm 1$ standard error across the different folds.

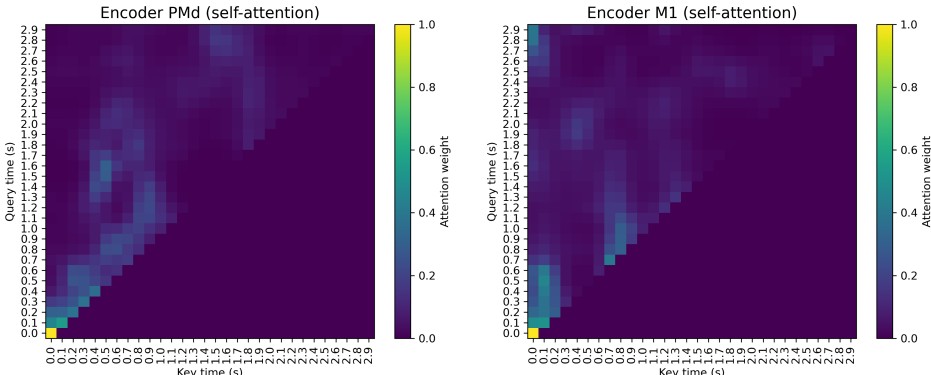

Figure 20: Self-Attention Maps obtained from CTAE's encoder branches

### G.3 ATTENTION MAPS

We note that as opposed to DLAG, our model does not learn latent-specific delays between brain regions. However, observing the attention maps indicates interesting temporal information.

In Fig. 20 we visualize the attention maps for the causal self-attention of the encoders for M1 and PMd separately, averaged over all trials and conditions. The encoders demonstrate the latent were closely aligned in time to both regions up to the "go" cue, after which they diverged. A diagonal feature in both attention maps, shifted with respect to each other, suggests a lag between the two regions following the cue.

In Fig. 21 we visualize the attention maps for the cross-attention in the decoder encoders for M1 and PMd separately, averaged over all trials and conditions. In PMd the attention map shows a strong separation between the time periods before and after the cue. The interpretation of the M1 decoder attention map is less clear, without a clear separation of before and after the cue.

## H COMPARISON TO DEEPCCA

We applied DeepCCA on the M1-PMd and SC-ALM dataset and extracted shared latents between the pairs of regions. We implemented DeepCCA using parallel multilayer perceptrons (MLPs) for each regions inputs. Each branch consisted of three fully connected layers with 1024 hidden units, sigmoid nonlinearities, and batch normalization (non-affine) applied after each layer. The final layer applied batch normalization followed by a linear projection into a shared latent space that was matched to

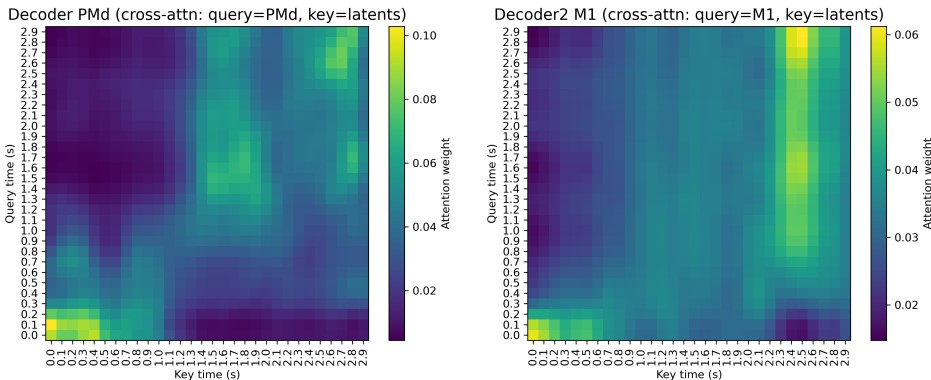

Figure 21: Cross-Attention Maps obtained from CTAE's decoder branches

that of the CTAE's latent dimensionality. Canonical correlation analysis (CCA) was then performed between the two branches to maximize shared variance in this latent representation. We evaluated these latents on the downstream tasks described in Sec. 5.1 and Sec. 5.2.

## H.1 M1-PMD DATA

The M1 branch received 66-dimensional inputs and the PMd branch 52-dimensional inputs.

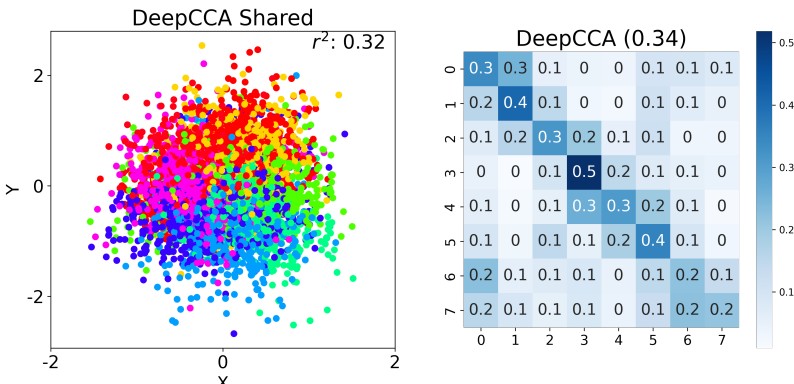

Figure 22: DeepCCA Decoding Performance on M1-PMd dataset. (a) Scatter plot of predicted hand-positions obtained from linear decoding from DeepCCA shared latents. (b) Confusion matrix for reach-direction classification at a single-trial level.

**Continuous Decoding.** As shown in Fig. 22 a), linear regression readout of the hand position from the shared latents, DeepCCA attained an $R^2$ value of $0.32$. In comparison, the shared latents obtained from CTAE and DLAG have $R^2$ values of $0.77$ and $0.72$. These results indicate the CTAE captures substantially more temporal variation than DeepCCA.

**Discrete Decoding.** Decoding the hand-reach condition for each trial using the shared latents, DeepCCA obtained an accuracy of $34\%$ at a single time-point level (Fig. 22 b)), compared to CTAE ($69\%$) and DLAG ($44\%$).

## H.2 SC-ALM DATA

Since DeepCCA performs pairwise correlation maximization, we applied it to region pairs (i.e.) SupSC-DeepSC, DeepSC-ALM and SupSC-ALM pairs. As in H.1, the input dimensions for each branch were matched to the dimensions of the number of neurons in each region.

Table 4: DeepCCA Pairwise Results on SC-ALM.

| Data | DeepCCA | CTAE |
|------|---------|------|
| SupSC-DeepSC | 0.22 | 0.59 |
| DeepSC-ALM | 0.22 | 0.59 |
| SupSC-ALM | 0.20 | 0.39 |

These results, shown in Table. 4, highlight DeepCCA's inability to model temporal dynamics, which hinders time-resolved decoding and interpretability. On the other hand, CTAE combines strong continuous decoding, robust discrete classification at a single-time point resolution and structured disentanglement between shared and private latents.

## I  COMPARISON TO CONCATENATED-DATA BASELINE

As a baseline, we used our transformer autoencoder backbone to analyze the concatenated data across all the regions, without enforcing latent disentangling into shared and private. As a result, we only use the orthogonality loss (Eq. 11) and the reconstruction loss (Eq. 8, across all regions) to train these models. The extracted latents were evaluated on both discrete and continuous behavioral decoding tasks. While this experiment provides a baseline to compare the amount of task-relevant information present in these latents, it is important to note that this baseline does not distinguish between computations that are performed in individual regions, or shared across different regions, as the recovered latents may be a mixture of both.

**M1-PMd Data.**  The CTAE model consistently outperformed the concatenated baseline model on both the continuous position and discrete condition decoding task (Fig. 23). On the continuous position decoding task, the concatenated baseline model achieves an $R^2$ of 0.59, compared to the $R^2$ of 0.77 and 0.72 obtained from CTAE and DLAG. The classification accuracy of the reach conditions obtained from the concatenated-data model is $57\%$, compared to accuracies of $69\%$ and $44\%$ obtained from CTAE and DLAG, respectively.

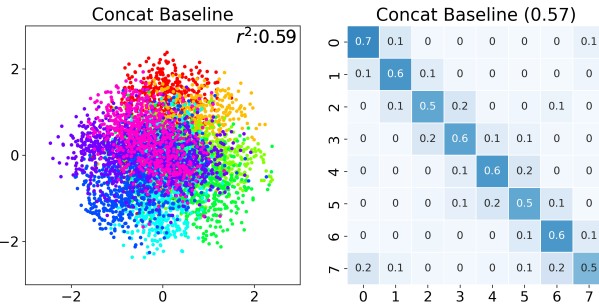

Figure 23: Concatenated-Data Decoding Performance on M1-PMd dataset. (a) Scatter plot of predicted hand-positions obtained from linear decoding from Concatenated-Data latents. (b) Confusion matrix for reach-direction classification at a single-trial level. True and predicted reach conditions on the Y- and X- axis respectively.

This highlights that the shared latents between M1-PMd, extracted from CTAE contain more task-related information than the latents obtained from concatenating the data.

**SC-ALM Data.**  The concatenated model outperforms CTAE in downstream behavioral decoding tasks of discrete modality classification (Fig. 24a) and continuous time decoding of target side (Fig. 24b). It is important to note that the latents obtained in this baseline do not identify computations that are region-specific and shared across regions. This is evident in (Fig. 24b) where the continuous-time decoding traces lacked the structure that is present in (Fig. 18).

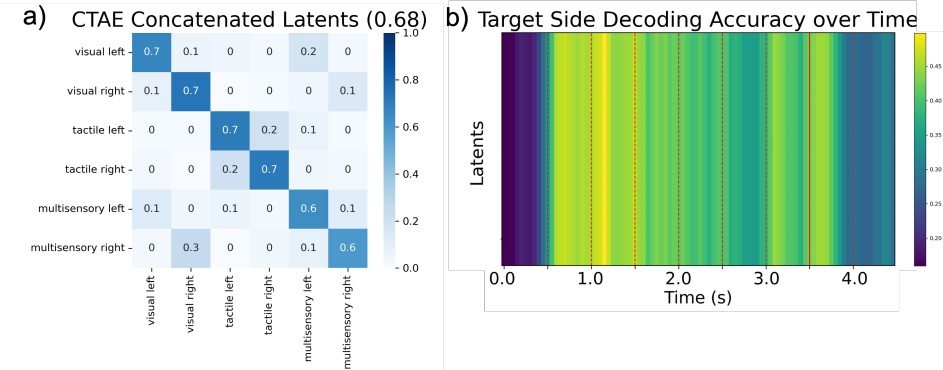

Figure 24: Concatenated-Data Decoding Performance on SC-ALM dataset. (a) Confusion matrix for task-modalities single-trial level. True and predicted modalities on the Y- and X- axis respectively. (b) Continuous-time target side decoding from the latents obtained from the Concatenated-data latents.

Thus, while the concatenated model can reflect dominant task information, it provides limited interpretability into the organization of region-specific versus shared computations. The concatenated approach collapses population activity into a single latent space, capturing task-relevant information, but not distinguishing whether that variance is shared across regions or specific to one region. In contrast, CTAE is designed to explicitly separate shared and region-private components of the neural activity. Thus, the two approaches reflect different conceptual goals—task-focused compression versus structured disentanglement—and their outputs are not directly interchangeable.

## J    THE USE OF LARGE LANGUAGE MODELS

We used ChatGPT solely for language-related assistance, such as improving grammar, refining phrasing, and polishing the presentation of the text. All research ideas, methods, and results are entirely our own. We take full responsibility for the content of this work.

## K    REPRODUCIBILITY STATEMENT

We provide all details necessary to reproduce our results.

- Model and objective: Sec. 4 describes the architecture and all loss terms, with explicit equations for reconstruction, shared-only reconstruction, alignment, and orthogonality (Eqs. 8–12). Multi-region extension: App. B specifies the $R > 2$ formulation, including masked latent fusion (Eq. 13) and the generalized training losses (Eqs. 14–18).
- Training and hyperparameters: App. C reports the full grid (Table 2), selected settings per dataset (Table 3), and the 10k-epoch training selection protocol/warm-up schedule.
- Compute resources: (PyTorch/CUDA on Quadro RTX 8000 + HPC) are listed in App. C.
- Data and preprocessing: App. D details sources and processing (bin sizes, Gaussian smoothing), with task specifics and counts for M1–PMd and SC–ALM.
- Evaluation protocols: Sec. 5.1–5.2 define the continuous (hand-position) and discrete (target/condition) decoding tasks and present cross-validated metrics/visualizations; App. E.2 details ablations and reports 5-fold statistics (Table 1).
- Baselines: App. G documents the DeepCCA comparison and evaluation setup.

