# OpenReview forum: "Coupled Transformer Autoencoder for Disentangling Multi-Region Neural Latent Dynamics"
_ICLR.cc/2026/Conference — ICLR 2026 Poster_

### Official Review · Reviewer_TAyW · 2025-10-28

**Soundness:** 4
**Presentation:** 4
**Contribution:** 4
**Rating:** 6
**Confidence:** 4

**Summary:**

The authors present an transformer-based autoencoder framework for modeling multi-region neural population data. The framework provides mechanisms for separately modeling shared and private variability across brain regions. The authors fit their model on two datasets - motor regions in monkey and multisensory regions in mouse - and analyze the information contained in the resulting subspaces.

**Strengths:**

CTAE represents a step forward in the literature of multi-region neural modeling. This framework is powerful and flexible, easily extending to more than 2 regions without an explosion of parameters that other models face. There is additional flexibility in defining subsets of regions that should have shared latent spaces, elegantly encoded in a single matrix. The transformer architecture also naturally takes into account temporal delays between regions, which previous models struggled to do in a scalable manner. The fitting procedure uses standard approaches (Adam) without too many bespoke elements (just a warm-up schedule for the orthogonality loss).

I appreciate that the authors explored the use of this model in two very different datasets: monkey vs mouse, motor vs sensory/cognitive areas, 2 vs 3 regions, etc.

The writing and figures are very clear.

**Weaknesses:**

Results on real data are hard to interpret. For example, the authors state "...the shared PMd–M1 subspace captured the dominant task-relevant signals. In contrast, DLAG tended to assign behaviorally relevant variance to both PMd and M1 private subspaces, potentially due to less effective separation between shared and private dynamics." While this feels intuitively true, it's impossible to know which model is "more correct". The example with the second dataset is even more complex, and hard to if these results are "good" or not. I'm wondering if there are other manipulations that can further highlight the benefits of this model. For example, what if you fit the model on data concatenated across the regions, such that there is just a single set of latents, and then use these latents to decode continuous hand position and target condition. How do these values compare to the private/shared subspaces? This sort of analysis would represent a "baseline" that is perhaps easier to compare to than DLAG or other existing models.

Following the previous comment, this work could be strengthened considerably with simulated data. Can CTAE recover the ground truth when the model hyperparameters match those of the data generation process (e.g. number of latents)? How does performance change when the model hyperparameters are not matched (e.g. more/fewer latents)? How much data is actually needed to train the CTAE (under matched hyperparameters, say)? These questions should be explored more thoroughly with simulated data given the difficulty of interpreting the results on real data.

**Questions:**

L85: is "absent" supposed to be here?

The authors claim one advantage of their framework is "Generic downstream utility", i.e. that the latent space is behavior agnostic. However, a robust line of work is developing that attempts to partition neural latents into "behaviorally-relevant" and "behaviorally-irrelevant". While directly addressing this is outside the scope of this work, I'm curious if the authors think there is room (and/or utility) in expanding their approach to encompass this desire for partitioned subspaces.

Are the trials the same length in both datasets? If not, how do authors manage batches composed of different-length sequences during training?

One of the major benefits (in my opinion) of the transformer-based approach is that it automatically takes care of time delays between different regions, in a much more robust, adaptive, and elegant way than previous approaches. If this is indeed the case, the authors should highlight this advantage in the main text. How could one actually dive into this more deeply? If there is a way to infer trial-to-trial lags between regions that would be incredibly useful (and interesting to compare to the lags inferred by DLAG).

---

> ### Author Response · Authors · 2025-11-22
>
> We thank the reviewer for emphasizing CTAE's flexibility, its scalability to multi-region neural activity and the advantages offered by the transformer backbone in handling temporal structure. We also appreciate the positive feedback provided regarding the clarity of the manuscript and the detailed assessment, and insightful suggestions provided.
> ### Baseline Comparisons
> We thank the reviewer for pointing out this baseline. In the revision, we will include a single Transformer autoencoder trained on the concatenated activity from all regions. This model serves as a meaningful control but, by construction, cannot disentangle shared and private latents or capture region-specific timing differences. We will report both continuous and discrete decoding performance for this baseline and explicitly contrast its behavior with CTAE's structured disentanglement.
> ### Interpreting CTAE vs. DLAG on Real Data
> We agree that interpretation on real data can be challenging, as ground truth latent structure is unknown. Our goal in comparing CTAE with DLAG is not to claim a single "correct" solution, but to demonstrate interpretability and consistency with known neuroscience. In the M1-PMd reaching dataset, CTAE identifies a coherent shared subspace capturing the dominant task-relevant signals (e.g., hand-position dynamics). In contrast, DLAG tends to distribute behaviorally relevant variance across private PMd and private M1 latents in a directionally anisotropic manner.
> This is reflected in both:
> - **continuous decoding**: private latents predict certain movement directions but not others
> - **discrete target classification**: confusion matrices show high accuracy for only a subset of reach directions
>
> Such direction-specific "fragmentation" of kinematic information across private latents is inconsistent with established findings that **both PMd and M1 encode reach kinematics through a coherent, low-dimensional manifold**. This pattern suggests that DLAG may inadvertently leak shared information into private subspaces, whereas CTAE's mask structure and regularization prevent such leakage.
> ### Synthetic Data Experiments
> We thank the reviewer for highlighting the value of synthetic experiments for assessing ground-truth recovery and for guiding choices such as the number of latents and the amount of data required. We agree that such analyses would nicely complement our current results. At the same time, synthetic spike trains necessarily rely on simplified assumptions about the underlying latent dynamics and observation process, and therefore cannot fully capture the richness and biological complexity of real neural activity. As a result, any conclusions about model capacity or data requirements drawn from synthetic data should be viewed as suggestive rather than definitive. In the present work, we deliberately chose two widely used datasets spanning different circuits and behaviors to ground our results in realistic settings.That said, we are in the process of designing controlled simulated datasets to systematically evaluate CTAE's capacity. These experiments are ongoing, and we will include the resulting analyses in the final version of the manuscript.
> ### L85 - "absent"
> Thank you, this is a typo. We will correct it.
> ### Behavioral relevance of the latents
> We thank the reviewer for raising this interesting point.  CTAE in its current form is intentionally behaviour-agnostic, allowing the same latent space to support diverse downstream decoding tasks without supervision. That said, the architecture readily accommodates extensions in which supervised behavioural information is used to guide or partition the latent space-for example, via additional masks, contrastive objectives, or losses that encourage specific latent dimensions to be predictive (or invariant) with respect to behaviour.  We view integrating such behaviour-supervised structure into CTAE as a promising direction for future work, particularly for identifying task-relevant vs. task-irrelevant neural subspaces and will add this to the Conclusions in the revision.
> ### Trial lengths
> The trials in each dataset presented in Section 5, have fixed lengths. For the M1-PMd dataset, each trial contains 30 time bins (3 seconds at 100ms resolution). For the SC-ALM dataset, each trial contains 90 time bins (roughly 4.5 seconds at 50ms resolution). Because all trials within a dataset share the same length, batching does not require padding or masking for variable-length sequences. That said, since CTAE has  a transformer architecture, it can readily accommodate trials of varying lengths with padding and masking.
> ### Inter-region delays
> We thank the reviewer for this observation and for highlighting this important advantage of Transformer-based models. Because CTAE uses *causal* self-attention, each region's encoder can adaptively integrate information over its past, which implicitly accommodates inter-regional delays without requiring parametric lag terms.

---

> > ### Comment · Reviewer_TAyW · 2025-11-23
> >
> > I thank the reviewers for thoroughly addressing my comments. I have a few remaining comments:
> >
> > **CTAE vs DLAG**
> >
> > I found the explanation given here a bit more complete and convincing than what was in the initial submission, and encourage the authors to use their additional page of space for a final manuscript to describe the differences more thoroughly in the main text.
> >
> > **Synthetic data experiments**
> >
> > I whole-heartedly agree with the reviewers that simulated data will not contain the complexity of real neural data, and that successful CTAE inference on simulated data does not imply the model will learn the appropriate structure in real data. _However_, if the model fails to capture the simplified structure in simulated data then I certainly would not trust it to capture more complex structure in real data. I appreciate the effort of designing a good simulated data experiment, and look forward to the results!
> >
> > **Inter-region delays**
> >
> > Using a causal self-attention mechanism to remove the need for parametric lag terms is, in my opinion, one of the more powerful aspects of this model. I would be curious to see some simple analyses of lag between regions, perhaps by showing attention maps across trials - how variable are the lags between regions from trial to trial? A proof-of-concept for how to investigate this type of question with CTAE would strengthen the paper and be of considerable interest to many readers I think.

---

### Official Review · Reviewer_MYUp · 2025-10-29

**Soundness:** 4
**Presentation:** 4
**Contribution:** 3
**Rating:** 8
**Confidence:** 4

**Summary:**

The authors propose Coupled Transformer Autoencoder (CTAE) for modeling simultaneous recordings from multiple brain areas using transformer based encoder and decoder priors. CTAE can extract the shared and private representations between regions thus making it a more interpretable model for neuroscience to understand the interactions between different brain regions. The authors demonstrate the utility of CTAE on two electrophysiology datasets.

**Strengths:**

- CTAE uses a novel objective that explicitly encourages orthogonality between latent dimensions and also prevents region specific variance from leaking into the shared space. Compared to most of the other multi-region and multi-modal models in neuroscience, I like how principled the objective function has been constructed for a more clean way of ensuring the separation of shared and private latents.

- The introduction of a transformer-based autoencoder for multi-region neural recordings is a unique contribution as far as I know. And this can act as a more flexible prior that can capture the necessary long range temporal dependencies that previous GP or RNN based models might not be able to.

- CTAE is scalable to more than two regions, making it suitable for a variety of neuroscience datasets.

- Clear organization and presentation.

**Weaknesses:**

Although these are [1, 2,3,4] multi-modal models, I find them to be closely related in terms of capturing the interpretable shared and private latent representations in neuroscientific data. Did the authors consider comparisons to multi-modal models in neuroscience (e.g. modifying them to handle multi-region data for comparisons)?

1. Vahidi, P., Sani, O. G., & Shanechi, M. M. (2025). BRAID: input-driven nonlinear dynamical modeling of neural-behavioral data. arXiv preprint arXiv:2509.18627.
2. Schneider, S., Lee, J. H., & Mathis, M. W. (2023). Learnable latent embeddings for joint behavioural and neural analysis. Nature, 617(7960), 360-368.
3. Gondur, R., Sikandar, U. B., Schaffer, E., Aoi, M. C., & Keeley, S. L. Multi-modal Gaussian Process Variational Autoencoders for Neural and Behavioral Data. In The Twelfth International Conference on Learning Representations.
4. Yi, D., Dong, H., Higley, M. J., Churchland, A., & Saxena, S. Shared-AE: Automatic Identification of Shared Subspaces in High-dimensional Neural and Behavioral Activity. In The Thirteenth International Conference on Learning Representations.

**Questions:**

Please refer to the Weaknesses section.

---

> ### Author Response · Authors · 2025-11-22
>
> We thank the reviewer for acknowledging the principled construction of our objective functions, and the novelty and flexibility provided by our transformer architecture. We also appreciate your recognition of the scalability of CTAE and for your positive assessment of the manuscript’s clarity.
>
> ### Comparison to Multimodal Models
> We thank the reviewer for pointing us to these relevant works; we will cite them in the revision in Related work and explain how our approach differs from these. The cited works operate in settings fundamentally different from ours. BRAID models a single shared latent dynamical system and does not support private subspaces or more than one neural population. Schneider et al. is supervised (behavior $\rightarrow$ neural) and not a multimodal or multiregion model. Multimodal GP-VAE and Shared-AE scale poorly to $R>2$ regions due to exponential growth in experts or pairwise loss terms, and both operate on i.i.d. samples or short windows rather than full neural time series.
>
> In contrast, CTAE is designed specifically for **multi-region neural time-series** with disentangled shared, subset-shared, and private latent dynamics that scale without parameter explosion.

---

### Official Review · Reviewer_W2Xz · 2025-10-31

**Soundness:** 4
**Presentation:** 3
**Contribution:** 4
**Rating:** 8
**Confidence:** 4

**Summary:**

This paper addresses the need to have a model that can effectively disentangle the shared components that mediate inter-area interactions from private signals unique to each region for mechanistic insight and designing causal experiments. The authors present previous work and claim that single-area tools fail when they are applied to concatenations of recordings (essentially making the data multi-area). Furthermore, they also mention that recent efforts show a subset of latent dimensions actively participate in inter-area communication. This subspace is believed to be orthogonal to region-specific dynamics. Other CCA-based approaches treat timepoints as iid samples and discard dynamics. This shows the need for CTAE.
Recorded neural activity across 2 brain regions is modeled as a nonlinear function of underlying latent dynamics specific to each region.
S = shared dynamics,
P^1 = dynamics local to region 1,
P^2 = dynamics local to region 2

The model is a coupled autoencoder built on transformer blocks that disentangles shared and private representations. The loss function recovers latents that maximally represent the neural activity. Each encoder is a Transformer stack with self-attention layers that capture long-range, nonlinear temporal dependencies within a region. Each decoder employs standard Transformer cross-attention to reconstruct the original firing rates from its region’s latents.

Overall model (Figure 2):
2 regions of the brain have separate data. These 2 regions are assumed to be sending signals to each other. The data is smoothed and passed into its individual encoder. The encoder brings the dimensionality down to some inferred latent representation. Each of these latent rep vectors is multiplied by a weight mask, and the result is added to produce a single fused latent. This fused latent contains signals that are local to region1/2 and also shared across regions. To disentangle these signals, 2 separate decoders are used.

Loss: they use 4 loss functions.
Reconstruction loss: so that the autoencoder reproduces its own region’s activity.
Shared-only reconstruction loss such that each decoder reconstructs neural activity from its region using only the shared representation. The authors do this to prevent the decoder from shifting information into private subspaces, leading to inaccurate representations.
Alignment loss:  The shared latents are meant to capture only dynamics common to both regions. To enforce this, they align each encoder’s shared output to their average, ensuring consistency across regions and preventing region-specific variance from leaking into the shared space.
Orthogonality loss. To encourage each latent coordinate to capture distinct, non-redundant structure.

They apply CTAE on the dorsal premotor cortex (PMd) and the primary motor cortex (M1) in macaque monkeys performing a standard delayed center-out reaching task with eight outward targets.

They evaluate their inferred latents on 2 simple linear decoding tasks: predict hand position, and predict target (out of the 8 targets).
They hypothesize that the shared latent space would encode a majority of behaviorally relevant information, particularly target identity. M1 would be for finer temporal structure related to movement, and PMd is for higher-order planning signals.
They find: the shared latent subspace captured with CTAE showed dominant task-relevant signals. DLAG, in contrast, tended to assign this behavioral info to both PMd and M1 private subspaces.

For the 2nd test dataset, they use data from S,C which integrates visual and tactile information and contributes to orienting behavior and whereas ALM encodes preparatory and choice-
related activity during decision-making tasks. The shared subspace captures task-relevant features such as stimulus type and target side. They confirm these findings with CTAE, but they do not contrast it with any other method.

**Strengths:**

the method is simple and scalable to more than 2 regions. It is also potentially useful for other kinds of time series data outside of neuroscience. 1 experiment’s comparison with DLAG shows that CTAE is better. 2nd experiment compares CTAE to known region functions and proves that the output from the model matches expectations of region behavior.

**Weaknesses:**

1. comparison to other methods on only one dataset: For experiment 2 - the authors should have tested DLAG here as well.
2. The interpretation of the results could be improved.

**Questions:**

1. what is the meaning of different degrees of shared or private activity in each brain area, and should we be surprised when decoding a task-relevant variable succeeds in one case and fails in another?

---

> ### Author Response · Authors · 2025-11-22
>
> We thank the reviewer for highlighting the principled design of loss functions and the relevance of disentangling shared and private latents, for understanding inter-area interactions. We also appreciate your recognition of the scalability of CTAE to multiple regions and the broad applicability of CTAE. We address the comments below.
>
> ### Broader impact of CTAE
> We thank the reviewer for noting the point that CTAE’s design makes it broadly applicable to other types of multiview time-series data.  We agree with this observation and will add this to the conclusion section of the revised manuscript.
>
> ### Comparison to DLAG/mDLAG
> We agree with this suggestion. Since SC–ALM involves three regions,  we will add results using mDLAG (Gokcen et al.)  in the revised manuscript.
>
> ### Interpretation of Results
> We appreciate this feedback. In the revision, we will provide clearer neurophysiological interpretations of each shared/private subspace, to make the results easier to interpret while emphasizing the mechanistic insights gained through CTAE.
>
> ### Interpreting the shared and private activity
> **Shared subspaces** reflect neural dimensions that covary across regions and therefore typically encode *global behavioral signals* that are broadcast across the circuit.
>
> **Private subspaces** capture *locally generated computations*, idiosyncratic dynamics, and information irrelevant or orthogonal to the other region.
>
> Thus, strong decoding from shared latents indicates **distributed encoding** (e.g., target identity), while decoding from private latents highlights **local computations** (e.g., anticipatory PMd signals or ALM delay activity).

---

### Official Review · Reviewer_ULYq · 2025-11-01

**Soundness:** 3
**Presentation:** 3
**Contribution:** 3
**Rating:** 6
**Confidence:** 4

**Summary:**

This paper introduces a new transformer-based autoencoder for modeling neural data collected from multiple brain regions, called CTAE. It deals with the disentanglement of each region's private latent and the shared latent well, resulting in a more biologically interpretable latent subspace and dynamic results.

**Strengths:**

* The proposed method is intuitive and powerful.
* The experiments and theoretical analysis are comprehensive, especially for the method extension to multiple regions and the real-world experiments.

**Weaknesses:**

* There seems to be no synthetic experiment in the main paper.
* The reconstruction loss is MSE, which means a Gaussian distribution. This is a technically inappropriate distribution for Poisson spike data, as assumed by the authors in line 54. The real-world dataset is also Poisson spike count.

**Questions:**

* Why not directly require the shared latent subspace to be just one thing? What's the consideration of splitting them and then aligning them during training? Implementing such a model and other variants is easy and can be viewed as intermediate baselines.
* Is there any theoretical conclusion about the uniqueness of the discovered latent?
* For the reaching task dataset, what's the reason for assuming hand position is largely represented in the shared subspace?

---

> ### Author Response · Authors · 2025-11-22
>
> We thank the reviewer for recognizing the intuitiveness of CTAE and its scalability in disentangling shared and private latents across multiple brain regions. We appreciate your comments on the strengths of our experimental analysis.
> ### Synthetic Experiments
> We thank the reviewer for highlighting the value of synthetic experiments. We are designing datasets to evaluate CTAE and these experiments are ongoing. We will include the results in the final version of the manuscript.
>
> ### MSE in reconstruction loss
> In our existing analysis, the spike trains are **Gaussian-smoothed to obtain continuous firing-rate estimates** (illustrated in Figure 2), following standard practice in modeling motor-cortex and multisensory population activity. After smoothing, the inputs are **continuous-valued and approximately homoscedastic**, making MSE an appropriate and widely used choice for neural latent modeling (also adopted in, NDT, Deep Autoencoders for neural data, etc.).
>
> That said, we agree that likelihoods matched to spike generation (e.g., Poisson or negative-binomial) are preferable when operating directly on unsmoothed spike counts. **An exciting future direction is training CTAE directly on spikes with a Poisson/dispersion-aware observation model**, which our architecture readily accommodates but we did not pursue here to keep the focus on multi-region disentanglement.
>
> Thank you for bringing these important points to our attention. We will clarify in the revision (Section 3, page 3) that “the recorded spike trains are Gaussian-smoothed to obtain continuous firing-rate estimates”. Throughout the remainder of the paper, we will retain our existing terminology and consistently refer to these signals as continuous firing-rate estimates.
> ### Aligning Latents versus Enforcing a Single Shared Subspace
> A formulation that directly defines a single shared latent $S$ as a function of all inputs, e.g., $S = h(X^{(1)}, X^{(2)})$, does not scale when moving beyond two regions. In realistic multi-area recordings, shared structure is not only ``across all regions''; there exist pairwise-shared, triple-shared, and arbitrary-subset shared components. If the model must explicitly learn each shared latent as
> $$S_{\mathcal{A}} = h_{\mathcal{A}}\bigl(X^{(r)} : r \in \mathcal{A}\bigr)$$
> then every non-empty subset $\mathcal{A} \subseteq \{1,\dots,R\}$ would require its *own* encoder, resulting in an exponential number of encoders (up to $2^{R}-1$). This quickly becomes intractable even for moderate $R$.
>
> **To avoid this exponential blow-up**, we instead let each region learn its own estimate of the shared latent,
> $$S^{(1)} = h_1(X^{(1)}), \qquad S^{(2)} = h_2(X^{(2)})$$
> and enforce that these estimates *align* through (i) the masked fusion rule, (ii) the alignment loss, and (iii) the shared-only reconstruction loss. This strategy preserves region-specific coordinate systems, keeps the model scalable, and implicitly enables the discovery of all subset-shared components without requiring separate encoders for each subset.
>
> ### Uniqueness of Discovered Latents
> As with most latent variable models (e.g., DLAG, GPFA, multiview autoencoders), the latent coordinates are not uniquely identifiable: they are only determined up to an invertible linear transformation (e.g., rotation) within each latent block. What is typically identifiable is the **subspace** spanned by shared or private latents, rather than a unique basis for that subspace. In CTAE, we introduce several identifiability-promoting design choices that encourage a more stable decomposition into private, partially shared, and shared components:
> - Fixed mask structure for private, partially shared, and shared latents across regions (Eq.3), which constrains how each latent can load onto each area.
> - Regularization on the latent representation, including an orthogonality loss that discourages redundancy across latent dimensions, and a shared-only reconstruction objective that encourages the shared latents to capture as much cross-region structure as possible (Eqs. 8–12).
>
> These constraints do not provide a formal guarantee of unique latent coordinates, but they substantially improve the practical identifiability of the subspaces. Empirically, we observe stable and reproducible latent subspaces across multiple random initializations, and we will clarify this in the revised manuscript in Section 4.2 *Training Objective* (page 6).
>
> ### Behavioral Readout for Interpretation: Hand Position Over Time
> This is motivated by extensive prior work:
> - PMd and M1 both strongly represent reaching kinematics and target variables (Churchland & Shenoy 2024; Kaufman et al. 2014).
> - Communication-subspace analyses (Semedo et al. 2019) show that behaviorally relevant signals lie in shared PMd-M1 dimensions.
>
> Our results align with these established findings: the shared latent space captures the majority of reach-related information, while private latents reflect finer local computations.

---

### Meta-Review · Area_Chair_xpra · 2026-01-09

**Summary:**

This paper introduces the Coupled Transformer Autoencoder (CTAE) for modeling simultaneous neural recordings across multiple brain regions while separating shared versus region-specific (private) latent dynamics. CTAE combines transformer-based sequence modeling for long-range, non-linear temporal dependencies with an explicit latent partitioning (shared/private and subset-shared via masks) and accompanying losses (alignment, shared-only reconstruction, and orthogonality) to reduce leakage between subspaces. The approach is evaluated on two large-scale, multi-region electrophysiology datasets (macaque motor and mouse multisensory/cognitive circuits), and the authors report that CTAE learns representations with improved downstream decoding performance and interpretable shared/private structure relative to prior approaches.

Overall, the paper was positively reviewed for its clear motivation and strong evaluations. Reviewers emphasized the novelty and flexibility of using a transformer autoencoder for multi-region neural time series (MYUp, TAyW) and found the loss design principled for disentangling shared vs private structure (W2Xz, MYUp). Reviewers also highlighted scalability beyond two regions and the ability to encode subset-shared structure through a masking matrix (TAyW, ULYq). The multi-dataset validation across species and circuits was also viewed as a strength (TAyW, W2Xz).

The main concerns raised in the initial reviews were largely addressed in the rebuttal and revision plan. Multiple reviewers asked for experiments on synthetic data. Reviewer ULYq questioned the use of MSE reconstruction given spike-count statistics and asked for clarity on the observation model; the authors clarified that they operate on Gaussian-smoothed firing-rate estimates, making MSE appropriate, and committed to clarifying this in the paper. Several reviewers requested stronger baselines and clearer interpretation: TAyW suggested a concatenated-data transformer AE baseline and emphasized interpreting real-data findings without ground-truth; W2Xz requested additional comparisons for the second dataset; and MYUp asked for discussion/comparisons to related multimodal/shared-private latent models. The authors responded by adding related work, adding a concatenated Transformer AE control, adding DeepCCA experiments, and committing to additional comparisons (including mDLAG where feasible) and improved interpretive framing, which reviewers indicated were satisfactory (e.g., TAyW follow-up).

Based upon the positive reviewers and extended analyses provided during the rebuttal, I am favor of acceptance. The authors should make sure to incorporate the additional clarification and experiments conducted during the rebuttal in the final version.

**Reviewer Concerns:**

Most of the reviewer concerns were addressed by the rebuttal. However, one reviewer expressed that they want to make sure that the authors provide thorough explanation of the differences between CTAE vs DLAG, requested simulated data experiments, and wants to still see some simple analyses of lag between regions, perhaps by showing attention maps across trials - how variable are the lags between regions from trial to trial.

**Reviewer Scores:**

I think the reviewers would maintain their scores.

---

### Decision · Program_Chairs · 2026-01-26

Accept (Poster)